# On Average-Case Error Bounds for Kernel-Based Bayesian Quadrature

**Xu Cai**                                                    *caix@u.nus.edu*
*Department of Computer Science*
*National University of Singapore*

**Chi Thanh Lam**                                            *lamchithanh1997@gmail.com*
*Department of Computer Science*
*National University of Singapore*

**Jonathan Scarlett**                                        *scarlett@comp.nus.edu.sg*
*Department of Computer Science, Department of Mathematics, Institute of Data Science*
*National University of Singapore*

**Reviewed on OpenReview:** *https://openreview.net/forum?id=JJrKbq35l4*

## Abstract

In this paper, we study error bounds for *Bayesian quadrature* (BQ), with an emphasis on noisy settings, randomized algorithms, and average-case performance measures. We seek to approximate the integral of functions in a *reproducing kernel Hilbert space* (RKHS), particularly focusing on the Matérn-$\nu$ and squared exponential (SE) kernels, with samples from the function potentially being corrupted by Gaussian noise. We provide a two-step meta-algorithm that serves as a general tool for relating the average-case quadrature error with the $L^2$-function approximation error. When specialized to the Matérn kernel, we recover an existing near-optimal error rate while avoiding the existing method of repeatedly sampling points. When specialized to other settings, we obtain new average-case results for settings including the SE kernel with noise and the Matérn kernel with misspecification. Finally, we present algorithm-independent lower bounds via distinct proof techniques to the existing ones.

## 1 Introduction

The integration of black-box functions is a fundamental problem with numerous applications, with Bayesian inference being a prominent example. The method of *Bayesian quadrature* (BQ) (O'Hagan, 1991; Rasmussen & Ghahramani, 2003) has particularly gained popularity, adopting Bayesian modeling techniques to model the unknown function and reduce the required number of function evaluations. Mathematically, the goal is to approximate the quantity

$$I(f) = \int f(\mathbf{x})p(\mathbf{x})d\mathbf{x}, \tag{1}$$

where $p(\mathbf{x})$ is a known weighting function, but we only have black-box access to $f(\mathbf{x})$. It is common to study this problem for worst-case functions in a given class, and with noiseless function queries and deterministic algorithms (e.g., see (Novak, 1988; Kanagawa et al., 2016; Kanagawa & Hennig, 2019)). However, this is also substantial motivation to understand average-case performance measures, noisy queries, and randomized algorithms (e.g., see (Novak, 1988; Plaskota, 1996; Novak, 2016)).

As an example motivating noisy settings, when performing integrals in Bayesian inference, function evaluations themselves may be implemented using a randomized subroutine whose variations can be modeled by introducing noise terms. As a rather different example, in the same way that noisy Bayesian optimization

(BO) can be used to find maximal sensor readings in a sensor network (e.g., see (Krause & Ong, 2011)), noisy BQ methods could be used to find average (or weighted average) readings.

## 1.1 Overview of Contributions

In this paper, our focus is on kernel-based BQ methods, corresponding to functions lying in a *reproducing kernel Hilbert space* (RKHS). We focus on the widely-used Matérn and Squared Exponential (SE) kernels, given by (Rasmussen & Williams, 2006)

$$k_{\mathrm{M}}(\mathbf{x}, \mathbf{x}') = \frac{2^{1-\nu}}{\Gamma(\nu)} \left( \frac{\sqrt{2\nu} \, \|\mathbf{x} - \mathbf{x}'\|}{l} \right)^{\nu} J_{\nu} \left( \frac{\sqrt{2\nu} \, \|\mathbf{x} - \mathbf{x}'\|}{l} \right), \tag{2}$$

$$k_{\mathrm{SE}}(\mathbf{x}, \mathbf{x}') = \exp \left( - \frac{\|\mathbf{x} - \mathbf{x}'\|^2}{2l^2} \right), \tag{3}$$

where $J_{\nu}$ is the modified Bessel function, $\Gamma$ is the gamma function, $l$ is the length-scale, $\nu$ is the Matérn smoothness parameter, and $\|\cdot\|$ denotes the Euclidean norm. Our main contributions are outlined as follows.

**Meta-Algorithm.** Our main algorithm is a "meta-algorithm" that builds on an idea originally used for Sobolev (and other) functions in noiseless settings, and appears to be traced back to (Bakhvalov, 1959). The meta-algorithm runs any algorithm for estimating the function in $L^2$-norm using the first half of the samples, runs a Monte Carlo (MC) method using the remaining samples to estimate the residual, and then combines the two. We give a general theorem relating the $L^2$-error of the first step with the overall integration error, and show that this has broad implications beyond the settings for which the idea was used previously, as outlined below.

Similar ideas have also been proposed for variance reduction as *control variates* (Ripley, 2009), and recently cast into a general functional approximation problem as *control functionals* (CF) (Oates et al., 2017). A notable distinction between our algorithm and CF is that we consider the effect of the noise, different assumptions are placed on $p(\mathbf{x})$, and the specific function classes that we consider are different (Matérn and SE kernels). See Section 1.2 for further discussion.

**Resulting Upper Bounds.** Some specific applications of our general theorem are as follows:

- By using a maximum-variance algorithm from (Vakili et al., 2021a) in the first part, we establish that for the Matérn-$\nu$ kernel, one attains an order-optimal error bound of $\Theta(T^{-\frac{\nu}{d}-1} + \sigma T^{-\frac{1}{2}})$ (with time horizon $T$, noise variance $\sigma^2$, Matérn smoothness $\nu$, and dimension $d$). This scaling was previously derived in (Plaskota, 1996), but used a method based on repeatedly re-sampling a carefully-chosen set of points many times to reduce noise, which may be undesirable in practice.

- We additionally provide corollaries of our main theorem providing results that appear to be new (though related results with other settings or criteria do exist), including for the Matérn kernel with misspecified smoothness, and average-case performance for the SE kernel in both the noiseless and noisy settings.

**Algorithm-Independent Lower Bounds.** While our main contributions are those outlined above, we also provide some results regarding algorithm-independent lower bounds. For the Matérn kernel with noise, we provide an alternative proof of the $\Omega(T^{-\frac{\nu}{d}-1} + \sigma T^{-\frac{1}{2}})$ lower bound from (Plaskota, 1996) that establishes a single source of difficulty for both terms in the bound (which were previously handled separately). Moreover, we establish that the $\Omega(\sigma T^{-\frac{1}{2}})$ lower bound holds much more generally in noisy settings, including for the SE kernel.

Our results are summarized in Table 1, along with various existing results that we discuss in the following subsection. Following the above outline, the paper structure is briefly described as follows: We formally introduce the problem in Section 2, state lower bounds in Section 3, provide the algorithm and its resulting upper bounds in Section 4, and provide experimental results in Section 5. Most technical proofs are deferred to the appendices.

Table 1: Summary of some of the most related existing results and our results. Results with an underline were partially or fully existing but are re-derived in our work (e.g., upper bounds based on distinct algorithms, or lower bounds via alternative proof techniques). Results with double underlines are new to this paper, to our knowledge.

| Summarized Results | Noiseless Worst | Noiseless Average | Noisy Average |
|---|---|---|---|
| **Matérn Lower** | $\Omega\big(T^{-\frac{\nu}{d}-\frac{1}{2}}\big)$ (Novak, 2016, Thm. 1) | $\Omega\big(T^{-\frac{\nu}{d}-1}\big)$ (Novak, 2016, Thm. 3) | $\Omega\big(T^{-\frac{\nu}{d}-1}+\sigma T^{-\frac{1}{2}}\big)$ (Plaskota, 1996, Thm. 4) |
| **Matérn Upper** | $O\big(T^{-\frac{\nu}{d}-\frac{1}{2}}\big)$ (Novak, 2016, Thm. 1) | $O\big(T^{-\frac{\nu}{d}-1}\big)$ (Novak, 2016, Thm. 3) | $O\big(T^{-\frac{\nu}{d}-1}+\sigma T^{-\frac{1}{2}}\big)$ (Plaskota, 1996, Thm. 4) |
| **SE Lower** | $\Omega\big(T^{-CT^{\frac{1}{d}}}\big)$ (Kuo et al., 2017, Thm. 1.1) (Linear algs. only) | N/A | $\Omega\big(\sigma T^{-\frac{1}{2}}\big)$ |
| **SE Upper** | $O\big(e^{-CT^{\frac{1}{d}}}\big)$ (Kanagawa & Hennig, 2019, Cor. 4.4) | $O\Big(T^{-\frac{C}{d}T^{-\frac{1}{d}}-\frac{1}{2}}\Big)$ | $O\big(e^{-C(\frac{T}{\log T})^{\frac{1}{d}}}T^{-\frac{1}{2}}+\sigma T^{-\frac{1}{2}}\big)$ |

## 1.2 Related Work

**Numerical Integration and Bayesian Quadrature.** Extensive early work on numerical integration appeared in the literature on information-based complexity, e.g., see (Bakhvalov, 1959; Novak, 1988; Traub, 2003; Novak & Woźniakowski, 2008) and the references therein. Function classes considered included Sobolev, Hölder, and others, with the Sobolev class being particularly related to the Matérn RKHS (see Appendix A). More explicit use of kernel-based methods appeared in (Narcowich & Ward, 2002; Wendland, 2004; Wendland & Rieger, 2005; Rieger & Zwicknagl, 2010), and Bayesian quadrature from a probabilistic perspective (Rasmussen & Williams, 2006; Kanagawa et al., 2016; 2018; Wynne et al., 2021) has gained particular popularity due to its role in statistical machine learning.

**Upper Bounds Under the Matérn Kernel.** For $d$-dimensional Matérn-$\nu$ RKHS functions with query budget $T$, early literature such as (Bakhvalov, 1959; Novak, 1988) proved that in the noiseless setting, the best possible worst-case (deterministic) error is $\Theta(T^{-\frac{\nu}{d}-\frac{1}{2}})$, whereas by considering the average-case error of a randomized algorithm, this can be reduced to $\Theta(T^{-\frac{\nu}{d}-1})$. An extensive survey of the noiseless setting can be found in (Novak, 1988), where it is also noted that basic MC sampling attains $O(T^{-\frac{1}{2}})$ error; this observation also extends immediately to the noisy setting with $\sigma = O(1)$ (Plaskota, 1996).

As we hinted above, the $L^2$-error bounds for the Matérn kernel with noise in (Wynne et al., 2021) will be particularly useful for our purposes, including in misspecified settings. Some implications of these results for noisy BQ were also noted in (Wynne et al., 2021), but they resulted in worse scaling (namely, $\Theta(T^{-\frac{s}{2s+d}})$ with $s = \nu + \frac{d}{2}$) than that shown in Table 1. In fairness, however, several of the results in (Wynne et al., 2021) can also be applied to settings with non-stochastic noise, whereas we only handle i.i.d. random noise.

**Upper Bounds Under the SE Kernel.** For functions in the SE RKHS, we observe in Table 1 that there are non-minor gaps between lower and upper bounds. For a different setting in $\mathbb{R}^d$ with $p(\mathbf{x})$ in (1) being a Gaussian distribution, (Kuo et al., 2017) gives a worst-case lower bound for *linear algorithms*, by decomposing SE functions using an orthogonal basis in the $L^2$ space. This lower bound has been further improved in (Karvonen et al., 2021) using another orthogonal basis in the native RKHS space, but it only applies to (linear) Gauss-Hermite rules. Overall, research on the SE kernel appears to have mainly focused on noiseless worst-case upper bounds, and exponential convergence has been developed via scattered data approximation (Wendland, 2004) and sampling inequalities for infinite smooth functions (Rieger & Zwicknagl, 2010).

A notable work related to ours is the adaptive BQ (ABQ) algorithm from (Kanagawa & Hennig, 2019). While their algorithm is adaptive, we see in Table 1 that our non-adaptive algorithm yields a slight improvement over theirs, albeit with a weaker average-case guarantee. Similar observations also apply to Matérn functions, where the ABQ algorithm only achieves $O(T^{-\frac{\nu}{d}})$ scaling (Kanagawa & Hennig, 2019, Cor. C.4), which falls short of the optimal scaling.

**Differences from control functionals.** As mentioned earlier, while the control functionals (CF) literature (e.g., Oates et al. (2017)) and our work share a similar approach in exploiting a tractable surrogate function with a bias-correction term in estimating the intractable integral, there are several differences between them:

- Our main focus is on the noisy setting, whereas CF algorithms are typically designed for noise-free scenarios.

- CF algorithms are typically developed under the restriction that $p(\mathbf{x})$ is intractable, so that pointwise access to the gradient information of $\log p(\mathbf{x})$ is necessary, for instance, by minimizing kernel Stein discrepancy using Stein operators (South et al., 2022). However, our setting assumes access to $p(\mathbf{x})$ and explores regret bounds accordingly.

- The focus of the CF literature has been accelerating the $O(\frac{1}{\sqrt{T}})$ convergence rate of MC or Quasi-MC (Oates & Girolami, 2016; Portier & Segers, 2019) under suitable assumptions, but to our knowledge less attention has been paid on attaining optimal or near-optimal convergence rates with matching lower bounds. We focus on this goal with an emphasis on the Matérn and SE kernels.

To highlight a representative example of the last dot point, we note that an $O(T^{-7/6})$ rate is established in (Oates et al., 2017, Thm. 3) under suitable assumptions on a class of gradient-based kernels, but this cannot encompass our noiseless results for the Matérn kernel in which, depending on the smoothness, it may be impossible to attain a rate of $O(T^{-7/6})$, or it may be possible to attain a much faster rate.

**Equivalence between Random Features and Bayesian Quadrature.** An interesting equivalence between worst-case kernel-based quadrature and random feature based function approximation in $L^2$ norm has once established in (Bach, 2017). While the main focus in (Bach, 2017) is the worst-case noiseless setting, it is noticed in Sec. 3.1 therein that their methods have a certain tolerance to noise. Noisy error bounds can be obtained via this fact, but they are higher than ours, with the first of the two terms (e.g., see (9) below) typically being a factor $\frac{1}{\sqrt{T}}$ smaller in our work, and being order-optimal in several cases of interest. We crucially rely on randomization to achieve this, whereas the proposed algorithm in (Bach, 2017) is deterministic. We note that for the SE kernel, one of our results (Corollary 4) uses a result from (Bach, 2017) as an intermediate step.

**Relationship with Bayesian Optimization.** The extensive literature on *Bayesian optimization* (BO) is also related to our work in the sense of iterating acquisition functions. However, BO turns out to be a strictly harder problem. As surveyed in detail in the noiseless setting in (Novak, 1988, Sec. 1.3), noiseless BO is closely related to estimating the function in $L^\infty$ norm, and noiseless BQ is closely related to estimating it in $L^2$ norm, with the former being strictly harder. Viewed differently, a key difficulty in BO is identifying a single short and narrow "bump" (Bull, 2011; Scarlett et al., 2017), whereas in BQ the same bump contributes a negligible amount to the integral. We elaborate on this connection in Appendix F, showing that BO-type techniques yield suboptimal BQ bounds for the Matérn kernel. On the other hand, in Section 4, we will see that this approach gives a fairly good result for the SE kernel in the noiseless (but not noisy) setting.

**Other Variants of BQ.** Finally, various works on BQ have explored more sophisticated techniques and variations such as adaptive sampling (Kanagawa & Hennig, 2019), active area search (Ma et al., 2014), and settings with multiple related functions (Gessner et al., 2019) (among many others). We are not aware of any (beyond those outlined above) that are directly related to our study of theoretical error bounds in relatively standard settings.

## 2  Problem Setup

Let $f : D \to \mathbb{R}$ be a real-valued function on the compact domain $D = [0, 1]^d$. By shifting and scaling, our results readily extend to arbitrary rectangular domains. In addition, our upper bounds easily extend to general compact domains.

We consider the class $\mathcal{H}_k(B)$ of functions whose RKHS norm $\| \cdot \|_{\mathcal{H}_k}$ is upper bounded by some constant $B > 0$. We focus in particular on the Matérn-$\nu$ kernel (see (2)), whose function class $\mathcal{H}_{\mathrm{M}}(B)$ is norm-

equivalent to the Sobolev class (see Appendix A for details). We also derive results for the SE kernel (see (3)), where the function class is denoted as $\mathcal{H}_{\mathrm{SE}}(B)$.

Let $p(\mathbf{x})$ be a known and bounded density function, i.e., $p(\mathbf{x}) \in [0, p_{\max}]$ for some $p_{\max} > 0$, and $\int p(\mathbf{x})d\mathbf{x} = 1$. We define $\mathcal{P}(p_{\max})$ to be the set of all functions satisfying these conditions. Our goal is to estimate the integral of an RKHS function $f : D \to \mathbb{R}$ weighted by $p(\mathbf{x})$, defined in (1). Before forming an approximation of this integral, the algorithm takes $T$ observations: At time step $t$, select $\mathbf{x}_t \in D$, and observe $y_t = f(\mathbf{x}_t) + \epsilon_t$, where $\epsilon_t \sim \mathcal{N}(0, \sigma^2)$ is Gaussian noise. The final approximate integral is denoted by $\hat{I}$.

As is commonly done for RKHS functions, we will use Bayesian methods based on a GP prior $\mathrm{GP}(0, k)$ and a Gaussian noise model. After observing $t$ noisy samples, the posterior distribution is also a GP with the following posterior mean and variance:

$$\mu_t(\mathbf{x}) = \mathbf{k}_t(\mathbf{x})^T \big(\mathbf{K}_t + \lambda_\mu \mathbf{I}_t\big)^{-1} y_t, \tag{4}$$

$$\sigma_t^2(\mathbf{x}) = k(\mathbf{x}, \mathbf{x}) - \mathbf{k}_t(\mathbf{x})^T \big(\mathbf{K}_t + \lambda_\sigma \mathbf{I}_t\big)^{-1} \mathbf{k}_t(\mathbf{x}), \tag{5}$$

where $y_t = [y_1, \ldots, y_t]^T$, $\mathbf{k}_t(\mathbf{x}) = \big[k(\mathbf{x}_i, \mathbf{x})\big]_{i=1}^t$, $\mathbf{K}_t = \big[k(\mathbf{x}_t, \mathbf{x}_{t'})\big]_{t,t'}$ is the kernel matrix, $\mathbf{I}_t$ is the identity matrix of dimension $t$, and $\lambda_\mu$ and $\lambda_\sigma$ are hyperparameters. Despite the Bayesian GP posterior needs $\lambda_\mu = \lambda_\sigma = \sigma^2$, as commonly accepted in recent RKHS-based BO literature, it is often useful to consider a "fictitious" GP posterior that the parameter $\lambda_\mu$ or $\lambda_\sigma$ may differ from $\sigma^2$. In this case, as revealed in (Kanagawa et al., 2018), the GP posterior mean is shown to be equivalent to the approximating function in *kernel ridge regression* (e.g., see (Caponnetto & De Vito, 2007)), where $\lambda_\mu$ is treated as the regularization parameter. On the other hand, when $\lambda_\sigma = 0$, the posterior variance is also known as the *power function* (Santin & Haasdonk, 2017), and will be frequently used in this paper. Nonetheless, we will continue to use the terms GP posterior mean and variance to refer to (4) and (5), respectively. Unless explicitly stated otherwise, when we mention $\lambda$, we consider a common choice $\lambda_\mu = \lambda_\sigma = \lambda$.

We consider both adaptive algorithms (i.e., the algorithm observes $y_1, \ldots, y_{t-1}$ before choosing $\mathbf{x}_t$) and non-adaptive algorithms (i.e., all $\mathbf{x}_1, \ldots, \mathbf{x}_T$ are chosen in advance). In fact, we will prove our lower bound for adaptive algorithms and upper bounds for non-adaptive algorithms, thus establishing the stronger type of result in both cases.

Using the shorthands $\mathcal{H}_k = \mathcal{H}_k(B)$ (as well as $\mathcal{H}_{\mathrm{M}}$ or $\mathcal{H}_{\mathrm{SE}}$ when the kernel $k$ is specialized as Matérn or SE) and $\mathcal{P} = \mathcal{P}(p_{\max})$, the settings in Table 1 are summarized as follows, the last of which is the one we primarily focus on:

- **Noiseless Worst-Case Error:** $\mathcal{E}_{\mathrm{wst}}^k(T) = \sup_{p \in \mathcal{P}, f \in \mathcal{H}_k} |I - \hat{I}|$.

- **Noiseless Average-Case Error:** $\mathcal{E}_{\mathrm{avg}}^k(T) = \sup_{p \in \mathcal{P}, f \in \mathcal{H}_k} \mathbb{E}\big[|I - \hat{I}|\big]$, where the expectation is over the randomized algorithm.

- **Noisy Average-Case Error:** $\mathcal{E}_{\mathrm{avg}}^k(T, \sigma) = \sup_{p \in \mathcal{P}, f \in \mathcal{H}_k} \mathbb{E}\big[|I - \hat{I}|\big]$, where the expectation is over the randomized algorithm and the noise.

We have omitted a notion of worst-case error for the noisy setting, since there are subtle issues in posing such a setting in a meaningful manner (e.g., see (Plaskota, 1996)). For instance, even if the algorithm is deterministic given the noisy observations, it can still obtain randomness by taking digits after the 1000th decimal point (say) of the observed values $y_t$. That is, the randomness from the noise alone could still give the same effect as using a randomized algorithm.

## 3 Lower Bounds

Our lower bounds for the noisy setting are stated as follows; note that we allow $\sigma$ to vary with $T$.

**Theorem 1.** (Average-Case Noisy Lower Bounds) *Consider our problem setup with constant parameters $(B, \nu, d, l)$, noise variance $\sigma^2$, and query budget $T$. Then, for any algorithm (possibly adaptive and/or randomized) for estimating $I$, we have the following lower bounds on the average-case error:*

---

**Algorithm 1** Two-batch integral estimation meta-algorithm

---

1: **Input:** Function $f$, domain $D$, time horizon $T$, function estimation algorithm ESTIMATEFUNC
2: Use ESTIMATEFUNC with $\frac{T}{2}$ samples to produce a function estimate $\hat{f}$
3: **for** $t = \frac{T}{2} + 1, \ldots, T$ **do**
4:      Sample $\mathbf{x}_t \sim p(\mathbf{x})$ independently
5:      Observe $y_t = f(\mathbf{x}_t) + \epsilon_t$
6: **end for**
7: Compute the approximate integral $\hat{I}_1 = \int_D p(\mathbf{x})\hat{f}(\mathbf{x})d\mathbf{x}$
8: Compute the residual $\hat{R} = \frac{2}{T}\sum_{t=T/2+1}^{T}(y_t - \hat{f}(\mathbf{x}_t))$
9: Output $\hat{I} = \hat{I}_1 + \hat{R}$

---

1. *For Matérn kernel with $\nu + \frac{d}{2} \geq 1$, we have*

$$\mathcal{E}_{\mathrm{avg}}^{\mathrm{M}}(T, \sigma) = \Omega\big(T^{-\frac{\nu}{d}-1} + \sigma T^{-\frac{1}{2}}\big). \tag{6}$$

2. *For any kernel such that $\max_{\mathbf{x}} k(\mathbf{x}, \mathbf{x}) < \infty$ and $\int_{[0,1]^d} k(\mathbf{x}, \mathbf{x}^\natural)d\mathbf{x} \neq 0$ for some $\mathbf{x}^\natural$ (e.g., the SE or Matérn kernel with $\mathbf{x}^\natural = \mathbf{0}$), we have:*

$$\mathcal{E}_{\mathrm{avg}}^{k}(T, \sigma) = \Omega\big(\sigma T^{-\frac{1}{2}}\big). \tag{7}$$

*Moreover, these lower bounds hold even under the fixed weight function $p(\mathbf{x}) = 1$.*

### 3.1 Proof of Theorem 1

The first term in (6) follows directly from noiseless average-case error bounds (see Appendix B). The second term can also be established by considering constant-valued functions (Plaskota, 1996), or as a special case of (7) (which is proved below). On the other hand, it is also of interest to establish a single hard subset of functions that yields both terms in (6) in a unified manner, thus establishing a single source of difficulty for both terms. We provide such an approach in Appendix D, considering functions composed of several small "bumps". The idea is that with too few samples the algorithm cannot reliably determine which bumps are positive and which are negative, and if too many of these are uncertain, then a certain level of error is unavoidable.

It remains to prove (7). To do so, we consider the function $f(\mathbf{x}) = k(\mathbf{x}, \mathbf{x}^\natural)$, which is bounded and has a non-zero integral by assumption. Moreover, since $k(\cdot, \mathbf{x}^\natural)$ is trivially in the RKHS class, we have that the scaled function $f_c(\mathbf{x}) = cf(\mathbf{x})$ has RKHS norm at most $B$ for all $c > 0$ below a suitably-chosen threshold.

We consider the sub-class of functions $f_c$ with RKHS norm at most $B$, and show that $\Omega(\sigma T^{-\frac{1}{2}})$ queries are needed even for this sub-class. To see this, we note that sampling any point $\mathbf{x}$ corresponds to observing $c$, scaled by $f(\mathbf{x})$, with $N(0, \sigma^2)$ noise. Without loss of optimality, we can assume that the algorithm only samples at the point(s) where $|f(\mathbf{x})|$ is largest (so that $c$ is maximally scaled while the noise level is fixed), and by assumption we have $|f(\mathbf{x})| < \infty$. Then, the noisy integration problem on this sub-class reduces to the noisy estimation of $c$ with Gaussian noise, which is well known to incur $\Theta(\sigma T^{-\frac{1}{2}})$ error (even when $c$ is known to be below an arbitrarily small constant threshold), e.g., see (Plaskota, 1996).

## 4 Upper Bounds

In this section, we introduce our main meta-algorithm and derive upper bounds on the average-case error. We follow the high-level idea of combining function estimation methods with Monte Carlo estimation on the residual, previously proposed for the noiseless setting (e.g., (Bakhvalov, 1959, Sec. 2) and (Novak, 2016, Thm. 3)), but with different details to account for the noise. The meta-algorithm is shown in Algorithm 1, and is described as follows. The samples are performed in two batches, but still in a non-adaptive manner

---

**Algorithm 2** Maximum variance sampling; can be used as ESTIMATEFUNC in Algorithm 1

---

1: **Input:** Function $f$, domain $D$, GP prior $\text{GP}(0, k)$, GP noise parameters $\lambda_\mu$ and $\lambda_\sigma$, time horizon $\frac{T}{2}$
2: **for** $t = 1, \ldots, \frac{T}{2}$ **do**
3:     Select $\mathbf{x}_t = \arg\max_{\mathbf{x} \in D} \sigma_{t-1}(\mathbf{x})$
4:     Receive $y_t = f(\mathbf{x}_t) + \epsilon_t$
5:     Update $\sigma_t$ using $\mathbf{x}_1, \ldots, \mathbf{x}_t$
6: **end for**
7: Update $\mu_{T/2}(\mathbf{x})$ using $\mathbf{x}_1, \ldots, \mathbf{x}_{T/2}, y_1, \ldots, y_{T/2}$
8: Output $\mu_{T/2}$

---

(i.e., the second batch can be chosen without knowing the first batch).[1] The first batch uses $T/2$ samples to construct an approximation $\hat{f}$ of $f$, which forms an initial estimate $\hat{I}_1 = \int_D p(\mathbf{x})\hat{f}(\mathbf{x})d\mathbf{x}$. The subroutine ESTIMATEFUNC for doing so is kept general for now, but by default, we will use maximum variance sampling[2] as shown in Algorithm 2, with the estimated $\hat{f}$ being given by the GP posterior mean $\mu_{T/2}$. Note that these $T/2$ samples are non-adaptive because the posterior variance of a GP does not depend on any observations. The remaining $T/2$ samples estimates the residual $R$ between the difference of the true integral $I$ and this value $\hat{I}_1$:

$$R = I - \hat{I}_1 = \int_D p(\mathbf{x})(f(\mathbf{x}) - \hat{f}(\mathbf{x}))d\mathbf{x}.$$

Having fixed $\hat{f}$ and hence $\hat{I}_1$, we can refine our overall estimate of $I$ by estimating the residual $R$. To do so, we use the last $T/2$ samples to construct a Monte Carlo estimator $\hat{R}$ of $R$, i.e.,

$$\hat{R} = \frac{2}{T} \sum_{t=T/2+1}^{T} (y_t - \hat{f}(\mathbf{x}_t)),$$

in which each $\mathbf{x}_t$ is sampled from $p(\mathbf{x})$. Hence, the approximated integral is simply $\hat{I} = \hat{I}_1 + \hat{R}$.

In the following theorem, we establish the upper bound of the average-case integration error of Algorithm 1 in terms of the noisy $L^2$ distance between $f$ and $\hat{f}$, with an extra noise term. This theorem is general and can be applied to a variety of kernel choices, noise settings, misspecified settings, etc., as we will see below. In the following, all omitted proofs are deferred to Appendix E.

**Theorem 2.** (Quadrature-$L^2$ Relationship) *For any fixed $f \in \mathcal{H}_k$ and $p \in \mathcal{P}(p_{\max})$, consider running Algorithm 1 with $T$ observations being corrupted by Gaussian independent noises $\mathcal{N}(0, \sigma^2)$, and obtain the estimates $\hat{f}$ and $\hat{I}$. Then, the average error satisfies*

$$\mathbb{E}[|I - \hat{I}|] \leq \sqrt{2p_{\max}} T^{-\frac{1}{2}} \mathbb{E}[\|f - \hat{f}\|_{L^2}] + \sqrt{2}\sigma T^{-\frac{1}{2}}, \tag{8}$$

*where the expectation is averaged over the randomness of the points queried and the noise.*

For the Matérn kernel, building on the results of (Wynne et al., 2021, Thm. 4), we obtain the following two corollaries.

**Corollary 1.** (Average-Case Noisy Matérn Upper Bound) *For $f \in \mathcal{H}_{\text{M}}$, consider the setup of Theorem 2, where the subroutine ESTIMATEFUNC for producing $\hat{f}$ is maximum-variance sampling (Algorithm 2) with parameters, $\lambda_\sigma = 0$ and $\lambda_\mu = \Theta(T^{-\frac{\nu}{d}})$.[3] Then, the average-case integration error is upper bounded by*

$$\mathcal{E}_{\text{avg}}^{\text{M}}(T, \sigma) = O\left(T^{-\frac{\nu}{d}-1} + \sigma T^{-\frac{1}{2}}\right). \tag{9}$$

---

[1] Our main theorem remains true when the first batch consists of adaptive sampling, but all of our corollaries will be based on non-adaptive sampling.

[2] We focus on maximum variance sampling for concreteness, but our analysis and results are unchanged when, at time $t$, an arbitrary point satisfying $\sigma_{t-1}(\mathbf{x}) \geq \gamma\|\sigma_{t-1}\|_{L^\infty}$ is found for some $\gamma \in (0, 1)$ (e.g., $\gamma = \frac{1}{2}$). This requirement is potentially much easier to attain in practice, instead of insisting on the global maximum.

[3] We consider $\lambda_\sigma = 0$ in order to exploit an existing result that the resulting maximum-variance rule gives a quasi-uniform set of points. Alternatively, any other quasi-uniform set could be pre-specified, such as a uniform grid.

As discussed in Section 1.2, Corollary 1 matches the lower bound (Theorem 1) up to constant factors, and shows that the overall difficulty of noisy BQ is as roughly hard as either noiseless BQ or univariate Gaussian mean estimation with variance $\sigma^2$ (whichever is harder). Since the algorithm used is non-adaptive but the lower bound applies even to adaptive algorithms, we conclude that there is no adaptivity gap in terms of scaling laws in this case.

Similarly to prior works such as (Kanagawa et al., 2016; Teckentrup, 2020; Wynne et al., 2021), we now consider the setting in which the smoothness hyperparameter $\nu$ is unknown, and Algorithm 2 is accordingly modified as follows:

- Instead of assuming that $\nu$ is given (via the prior $k_0$), we assume that the algorithm is given a sequence of estimates $\{\hat{\nu}_t\}_{t=1}^T$. These are kept generic, but could correspond to estimates that are iteratively updated as more data is collected.

- When forming the GP posterior at time $t$, the parameter $\hat{\nu}_t$ is used.

Following (Wynne et al., 2021), given the time horizon $T$, we define $\nu^- = \inf_{t \le T} \hat{\nu}_t$ and $\nu^+ = \sup_{t \le T} \hat{\nu}_t$,[4] and we assume that the number of distinct values in $\{\hat{\nu}_t\}_{t=1}^T$ is upper bounded by fixed constant.

**Corollary 2.** (Average-Case Noisy Matérn Upper Bound – Misspecified Setting) *For $f \in \mathcal{H}_M$, consider the setup of Theorem 2, where the subroutine ESTIMATEFUNC for producing $\hat{f}$ is the above-described modification of maximum-variance sampling with parameters $\{\hat{\nu}_t\}_{t=1}^T$ satisfying $\nu^+ > \nu$, and with $\lambda_\sigma = 0$ and $\lambda_\mu = \Theta(T^{-\frac{\nu^-}{d}})$ . Then, the average-case integration error is upper bounded by*

$$\mathcal{E}_{\text{avg}}^M(T, \sigma) = O\Big(T^{-\frac{\min(\nu^-, \nu)}{d} - 1} + T^{-\frac{\nu + \nu^- - \nu^+}{d} - 1} + \sigma T^{-\frac{1}{2}}\Big). \tag{10}$$

As a special case of this result, when $\nu^- = \nu^+$ (and both are greater than $\nu$, in accordance with the assumption $\nu^+ > \nu$), the scaling precisely reduces to that of Corollary 1. Thus, interestingly, over-estimating the smoothness parameter is not harmful in terms of scaling laws; similar observations were made in prior works such as (Kanagawa et al., 2016) (Remark 2 therein). More broadly, Theorem 2 can be used to convert other $L^2$ guarantees from (Wynne et al., 2021) (or from other works) to BQ guarantees, but for brevity we only formally state Corollary 2.

We now turn to the SE kernel without noise, by utilizing results from kernel-based interpolation on cubes (Wendland, 2004; Rieger & Zwicknagl, 2010), and obtain the following corollary.

**Corollary 3.** (Average-Case Noiseless SE Upper Bound) *For $f \in \mathcal{H}_{\text{SE}}$, consider the setup of Theorem 2, where ESTIMATEFUNC is chosen to take samples on a uniformly-spaced grid and return $\mu_{T/2}$ with $\lambda = 0$. Then, the average-case error is upper bounded by*

$$\mathcal{E}_{\text{avg}}^{\text{SE}}(T) = O\Big(T^{-\frac{C_s}{d}T^{-\frac{1}{d}} - \frac{1}{2}}\Big), \tag{11}$$

*for some constant $C_s > 0$.*

The same result can also be obtained with exponentially small but non-zero $\lambda$ (see Appendix E.3), but we provide the formal statement with $\lambda = 0$ for simplicity.

For SE kernel with noise, we follow the use of random features (Rahimi & Recht, 2007). We specifically build on (Bach, 2017), which analyzes weighted random Fourier features from a function approximation point of view. The approximation requires sampling from a leverage function distribution based on an infinite-dimensional integral operator. With the help of their analysis, we obtain the following corollary.

**Corollary 4.** (Average-Case Noisy SE Upper Bound) *For $f \in \mathcal{H}_{\text{SE}}$, under the setup of Theorem 2, there exists a random feature sampling algorithm that produces an approximation $\hat{f}$ in Algorithm 1 such that the average-case error is*

$$\mathcal{E}_{\text{avg}}^{\text{SE}}(T, \sigma) = O\Big(e^{-C_r(\frac{T}{\log T})^{\frac{1}{d}}} T^{-\frac{1}{2}} + \sigma T^{-\frac{1}{2}}\Big)$$

---

[4]This can be generalized so that these definitions also restrict $t \ge N$ for some finite $N$, but we simply take $N = 1$ to reduce notation.

*for some constant $C_r > 0$.*

In Section 1, we highlighted that BQ is related to RKHS-based optimization problems. We briefly note that the techniques from such works can also be used to obtain BQ bounds via Algorithm 2 (using it for $T$ observations rather than only the first half) or sampling on a uniform grid, but the resulting scaling is typically highly suboptimal. Specifically, in Appendix F, we show that this approach leads to the following:

$$\mathcal{E}_{\text{avg}}^{\text{M}}(T, \sigma) = O\Big(T^{-\frac{\nu}{2\nu+d}}(\log T)^{\frac{2\nu}{4\nu+2d}} + \sigma T^{-\frac{\nu}{2\nu+d}}(\log T)^{\frac{4\nu+d}{4\nu+2d}}\Big), \tag{12}$$

$$\mathcal{E}_{\text{avg}}^{\text{M}}(T) = O\Big(T^{-\frac{\nu}{d}}\Big), \tag{13}$$

$$\mathcal{E}_{\text{avg}}^{\text{SE}}(T, \sigma) = O\Big((\log T)^{\frac{d+1}{2}}T^{-\frac{1}{2}} + \sigma(\log T)^{\frac{d+2}{2}}T^{-\frac{1}{2}}\Big), \tag{14}$$

$$\mathcal{E}_{\text{avg}}^{\text{SE}}(T) = O\Big(e^{-\frac{d}{2}T^{\frac{1}{d}}}\Big). \tag{15}$$

These four results correspond to the Matérn noisy/noiseless and SE noisy/noiseless settings respectively. Algorithm 2 can be used directly to obtain the first three results, whereas (15) is based on sampling on a uniform grid. While these bounds above are weaker than our earlier results, the noiseless SE case (15) gives almost as fast decay to zero as (11).

## 5 Experiments

In this section, we conduct simulation studies[5] to compare our two-batch algorithm with its component parts, maximum variance sampling (MVS) and Monte Carlo sampling (MC). These experiments serve to verify that Algorithm 1 can be effective in practice, but we will also discuss some potential gaps between the theory and practice. In particular, our goal is *not* to establish state-of-the-art practical performance.

We refer to Algorithm 2 as MVS-MAT or MVS-SE when the kernel is Matérn or SE, and similarly for Algorithm 1 with MVS-MC-MAT or MVS-MC-SE. To attain a better understanding of how MVS-MC performs with respect to time, we modify it to *alternate* between MVS samples and MC samples, instead of doing all of one followed by all of the other. Mathematically, this does not change the behavior at the *final* time step.

### 5.1 Setup

**GP model.** We adopt the common choice $\nu = 3/2$ for Matérn-$\nu$ kernel, and the GP mean noise hyperparameter $\lambda_\mu$ is fixed as $10^{-5}$ for both kernels [6]. The lengthscale is left as a free parameter, as is an additional scale parameter that we introduce (multiplying (2)) to permit functions with varying ranges. Except where stated otherwise, these two parameters are learned by maximizing the data log-likelihood (Rasmussen & Williams, 2006) using the built-in SciPy optimizer based on L-BFGS-B, which is also used for finding the maximum variance point. We seek to solve the BQ problem with a constant weight function, i.e., $p(\mathbf{x}) = 1$.

**Evaluation.** We compare MVS-MC against its two components, MVS and Monte Carlo. For MVS and MVS-MC, we perform 100 trials, with each trial using a distinct set of 3 random initial points (but these are common to both methods). For MC, since every round is already randomized, we simply run 100 trials without initial points. For all functions, we consider a time horizon of $T = 250$, and evaluate the performance using the mean absolute error, with the ground truth value (and also $\hat{I}_1$ at Line 7 of Algorithm 1) being determined by trapezoidal rule with $10^5$ uniformly-spaced grid points (without noise). Errors are plotted with logarithmic scale for the purpose of more easily distinguishing values close to zero. Error bars in our plots indicate $\pm 0.5$ standard deviation with respect to the 100 trials.

---

[5] The code can be found at `https://github.com/caitree/Kernelized-Bayesian-Quadrature`.

[6] Our theory uses choices such as $\lambda_\mu = \Theta(T^{-\frac{\nu}{d}})$ with unspecified implied constants, which makes it difficult to choose the parameter in a way that exactly matches the theory. However, since the theory dictates choosing $\lambda_\mu$ to be small, we set it to be small here accordingly.

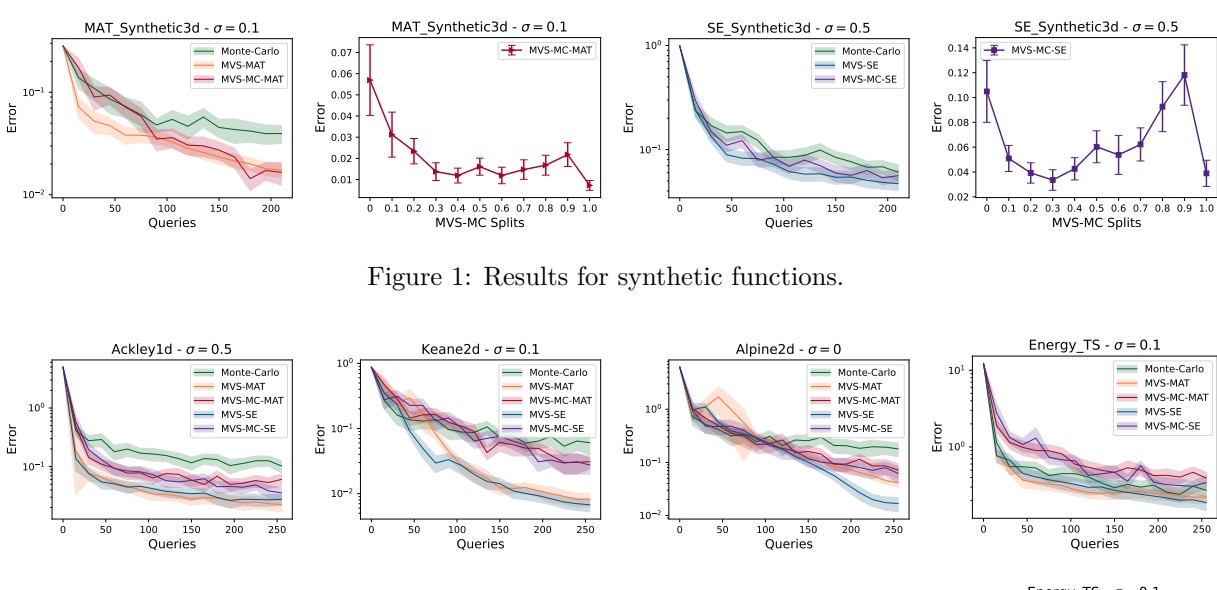

Figure 1: Results for synthetic functions.

Figure 2: Results for benchmark and real data functions.

**Varying the split size.** In our theoretical analysis, we let MVS-MC use half of the samples for each component (MVS and MC). However, in practice, it may be beneficial to allow different splits. In our experiments, we additionally explore the effect of the split fraction, i.e., the fraction allocated to the MVS part, and plot the final error with respect to that fraction. A split of 0 corresponds to MC, and a split of 1 corresponds to MVS. For these plots we use a linear y-axis scale, since the values are mostly further from zero and easier to compare on such a scale.

## 5.2 Results

We provide a selection of our results here, and give additional results in Appendix G.

### 5.2.1 Synthetic Kernel-Based Functions

We first simulate on a set of synthetic functions following the method of (Janz et al., 2020), where each function is constructed by sampling $m = 30d$ points, $\hat{\mathbf{x}}_1 \ldots \hat{\mathbf{x}}_m$, uniformly on $[0,1]^d$, and $\hat{a}_1 \ldots \hat{a}_m$ uniformly on $[-1, 1]$. The function is then defined as $f(\mathbf{x}) = \sum_{i=1}^{m} \hat{a}_i k(\hat{\mathbf{x}}_i, \mathbf{x})$. The length-scale and $\nu$ (in the case of the Matérn kernel) are set to be fixed as $0.2$ and $3/2$ respectively.

We let MVS and MVS-MC know the kernel hyperparameters exactly (i.e., they match the ones used to produce the functions). Some results for $d = 3$ and $\sigma \in \{0.1, 0.5\}$ are shown in Figure 1, and further $(d, \sigma)$ pairs are shown in Appendix G. We observe that MC performs well at the higher noise level, but performs poorly at the lower noise level, which aligns with the theory. At the final time step, MVS in fact performs well in both cases, though it can perform poorly in the low-query regime (e.g., Matérn 4D). While MVS-MC is not universally best, it is generally able to capture the benefits of both MVS and MC in this experiment.

### 5.2.2 Benchmark Functions and Sensor Measurement Function

We consider a variety of well-known benchmark functions, namely, Ackley, Alpine, Gramacy-Lee, Griewank and Keane; see (Bingham, 2013) for the descriptions. We also consider an experiment for sensor measurement data, which is described in Appendix G. A small selection of the results are shown in Figure 2, and the rest are shown in Appendix G.

These results again indicate that MC is better suited to higher noise levels, but they provide a somewhat less clear picture for MVS and the ideal MVS-MC split. In general, the error as a function of the split fraction can increase, decrease, or exhibit "U-shaped" behavior, though we again found the choice of 0.5 to usually be a good one (even if not always optimal).

Notably, in both the synthetic and benchmark experiments, the error can drop suddenly at 1.0, as this is where the MC component is no longer used and MVS-MC simply becomes MVS. This indicates that the theoretical benefits of MVS-MC over MVS may not always be observed in practice (e.g., because the theory hides important constant factors, or only captures the behavior for very large $T$). It may be of interest to seek refined theory addressing these findings in future work.

## 6 Conclusion

We have developed a framework for relating average-case quadrature error with $L^2$-function approximation error, allowing us to derive a number of both existing and new results. In addition, we explored algorithm-independent lower bounds with greater generality and/or distinct proofs compared to existing ones. In future work, it may be of interest to address some of the remaining gaps, such as those between the upper and lower bounds for the SE kernel (see Table 1).

### Acknowledgments

This work was supported by the Singapore National Research Foundation (NRF) under grant number R-252-000-A74-281. C. T. Lam was supported by the Singapore-MIT Alliance for Research and Technology (SMART) PhD Fellowship.

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

# A Connections Between Matérn RKHS and Sobolev Class

We briefly summarize some definitions and properties of the Sobolev function class, as stated and used in the work (Wynne et al., 2021) that we will later build on.

For integer-valued $s$, the Sobolev function class of order $s$ on a domain $D$ is defined as

$$W_2^s(D) = \Big\{ f \in L^2(D) : \sum_{\boldsymbol{\alpha}\,:\,|\boldsymbol{\alpha}|\leq s} \|\partial^{\boldsymbol{\alpha}} f\|_{L^2} < \infty \Big\}, \tag{16}$$

and the associated norm is given by

$$\|f\|_{W_2^s(D)} = \Big( \sum_{|\boldsymbol{\alpha}|\leq s} \|\partial^{\boldsymbol{\alpha}} f\|_{L^2}^2 \Big)^{1/2}, \tag{17}$$

where each $\boldsymbol{\alpha}$ is a $d$-dimensional multi-index with $|\boldsymbol{\alpha}| = \sum_{i=1}^d \alpha_i$, and $\partial^{\boldsymbol{\alpha}} f(\mathbf{x})$ is the weak derivative of order $\boldsymbol{\alpha}$. For a fractional (i.e., non-integer) order $s \in \mathbb{R}$ with $s \geq \frac{d}{2}$ (i.e., $\nu > 0$), the Sobolev function class of order $s$ is defined on $\mathbb{R}^d$ as

$$W_2^s(\mathbb{R}^d) = \Big\{ f \in L^2(\mathbb{R}^d) : \int_{\mathbb{R}^d} (1 + \|\mathbf{x}\|_2^2)^s |\hat{f}(\boldsymbol{\xi})|^2 d\mathbf{x} < \infty \Big\}, \tag{18}$$

where $\hat{f}(\boldsymbol{\xi})$ is the Fourier transform of $f(\mathbf{x})$. The associated norm is given by

$$\|f\|_{W_2^s(\mathbb{R}^d)} = \Big( \int_{\mathbb{R}^d} (1 + \|\mathbf{x}\|_2^2)^s |\hat{f}(\boldsymbol{\xi})|^2 d\mathbf{x} \Big)^{1/2}.$$

For functions defined over $D \subseteq \mathbb{R}^d$, the fractional Sobolev norm is taken as the infimum over all functions on $\mathbb{R}^d$ that match the original function when restricted to $D$:

$$\|f\|_{W_2^s(D)} = \inf \Big\{ \|f'\|_{W_2^s(\mathbb{R}^d)} : f' \in W_2^s(\mathbb{R}^d) \text{ and } f'(\mathbf{x}) = f(\mathbf{x}) \; \forall \mathbf{x} \in D \Big\}. \tag{19}$$

We note that we are slightly abusing notation here, since in the integer case (19) only matches (17) up to a constant factor (rather than exactly), but this will not impact any of our results in which the constant factors are kept implicit.

In the following, we will use $W_2^s$ as a shorthand for $W_2^s(D)$. Here we state a known result that characterizes the equivalence between Sobolev space and RKHS of Matérn kernels.

**Lemma 1.** (Sobolev Space & RKHS of Matérn kernels (Teckentrup, 2020, Prop. 3.3)) *Let $\mathcal{H}_M$ be the RKHS of the Matérn-$\nu$ kernel on $D = [0,1]^d$ (with any fixed positive length scale), and let $s = \nu + d/2$. Then, the Matérn RKHS is norm-equivalent to the Sobolev space $W_2^s$. That is, $\mathcal{H}_M = W_2^s$, and there exist constants $c_1, c_2 > 0$ such that for any $f \in \mathcal{H}_M$ (or equivalently, $f \in W_2^s$) we have*

$$c_1 \|f\|_{W_2^s} \leq \|f\|_{\mathcal{H}_M} \leq c_2 \|f\|_{W_2^s}. \tag{20}$$

# B Existing Results for the Noiseless Setting

The lower bound for the noiseless setting (i.e., $\sigma^2 = 0$) presented below restates the result of (Novak, 2016, Thm. 1 & 3), with a slight generalization to fractional Sobolev spaces where $s = \nu + \frac{d}{2}$ may be non-integer. Here and subsequently, the kernel parameters $\nu, l$, dimension $d$, and RKHS norm $B$ are all treated as constants, and we consider the limit $T \to \infty$.

**Theorem 3.** (Noiseless Lower Bound) *Consider the noiseless problem setup with constant parameters $(B, \nu, d, l)$ satisfying $\nu + \frac{d}{2} \geq 1$, and time horizon $T$. Then, we have the following lower bounds for the Matérn kernel in the noiseless setting:*

1. *For any algorithm (possibly adaptive), the worst-case error satisfies*

$$\mathcal{E}_{\mathrm{wst}}^{\mathrm{M}}(T) = \Omega(T^{-\frac{\nu}{d}-\frac{1}{2}}).$$

2. *For any algorithm (possibly adaptive), the average-case error satisfies*

$$\mathcal{E}_{\mathrm{avg}}^{\mathrm{M}}(T) = \Omega(T^{-\frac{\nu}{d}-1}).$$

*Moreover, these lower bounds hold even under the fixed weight function $p(\mathbf{x}) = 1$.*

While Theorem 3 is already known for Sobolev spaces with integer-valued $s$, fractional bounds are less commonly stated in early literature, though interpolation techniques exist for generalizing integer results to fractional ones (see, e.g. (Ritter, 2000, Prop. 8)). Overall, we believe it is useful to present a self-contained proof for the purpose of (i) completeness in handling the non-integer case, and (ii) introducing tools that will be re-used in the noisy setting (Appendix D).

These scaling laws are known to be order-optimal, since matching upper bounds (using non-adaptive algorithms) have been established, e.g., see (Novak, 1988, Sec. 1 & 3) for a detailed summary. For the average-case criterion, see also Corollary 1 with $\sigma = 0$.

## C    Proof of Theorem 3 (Noiseless Lower Bounds)

We generally follow the analysis of (Novak, 2016, Thm. 1 & 3), while adapting the notation to match ours, and making some minor adjustments in the analysis.

### C.1    Function Class Construction for the Matérn Kernel

We first describe a bounded-support class of functions consisting of multiple bumps. The following lemma introduces the associated function and some useful properties.

**Lemma 2.** (Bounded-Support Function Construction (Bull, 2011, Lem. 5), (Cai & Scarlett, 2021, Lem. 4)) *Let $h(\mathbf{x}) = \exp\left(\frac{-1}{1-\|\mathbf{x}\|^2}\right)\mathbb{1}\{\|\mathbf{x}\|_2 < 1\}$ be the $d$-dimensional bump function, and let $g_0(\mathbf{x}) = \frac{\epsilon}{h(\mathbf{0})}h\left(\frac{2\mathbf{x}}{w}\right)$, i.e., a rescaled version of $h$ for some $w > 0$ and $\epsilon > 0$. Then, $g_0$ satisfies the following properties:*

- $g_0(\mathbf{x}) = 0$ *for all $\mathbf{x}$ outside the $\ell_2$-ball of radius $w$ centered at the origin;*
- $g_0(\mathbf{x}) \in [0, \epsilon]$ *for all $\mathbf{x}$, and $g_0(\mathbf{0}) = \epsilon$.*
- $\|g_0\|_{\mathcal{H}_{\mathrm{M}}} \leq c_1 \frac{\epsilon}{h(\mathbf{0})}\left(\frac{1}{w}\right)^{\nu}\|h\|_{\mathcal{H}_{\mathrm{M}}}$ *when $k$ is the Matérn-$\nu$ kernel on $\mathbb{R}^d$, where $c_1$ is constant.*

In accordance with this lemma, let $g_0(\mathbf{x}) = \frac{\epsilon}{h(\mathbf{0})}h\left(\frac{2\mathbf{x}}{w}\right)$ be the rescaled bump function with values in $[0, \epsilon]$ and radius $\frac{w}{2}$ (i.e., diameter $w$). For suitably-chosen $M$, we consider $M$ such bumps with disjoint supports by shifting, and accordingly consider $2^M$ functions, one for each possible sign pattern of the bumps. Mathematically, all functions of the form $f(\mathbf{x}) := \sum_{i=1}^{M} \delta_i g_i(\mathbf{x})$ are contained in $\mathcal{H}_{\mathrm{M}}$, where $\delta_i \in \{+1, -1\}$ and $g_i(\mathbf{x})$ is the shifted version of $g_0(\mathbf{x})$. Since the bumps form a $d$-dimensional grid of step size $w$ in each dimension and the domain is $D = [0, 1]^d$, we have

$$M = \left\lfloor \frac{1}{w} \right\rfloor^d. \tag{21}$$

In fact, since the bumps have spherical support instead of rectangular, we could fit more of them into $[0, 1]^d$, but doing so would only impact the constant factors, which we do not seek to optimize.

A 1D example of $f(x)$ is illustrated in Figure 3.

Letting $I_0 := \int_{\|\mathbf{x}\| \leq \frac{w}{2}} g_0(\mathbf{x})d\mathbf{x}$ denote the integral of $g_0$, we have

$$I_0 \geq \Omega(w^d \epsilon), \tag{22}$$

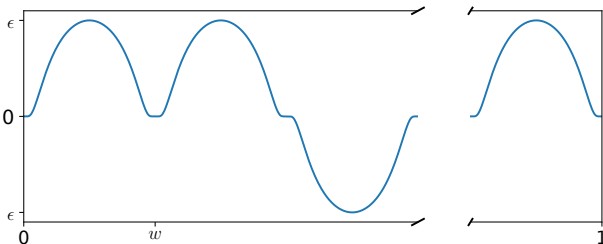

Figure 3: 1D illustration of a function in the restricted Matérn function class.

which follows from the fact that the integral of $h(\mathbf{x})$ is constant, the vertical scaling shrinks the integral by $\epsilon$, and the horizontal scaling shrinks the integral by $\Theta(w^d)$.

To evaluate the RKHS norm of each $f$, we use the equivalence of the RKHS norm of $f$ and the Sobolev norm, as stated in Lemma 1. Since each $f$ is a sum of $M$ disjoint-support functions, we can define these disjoint regions as $D_1, \ldots, D_M$. First considering the case of integer-valued $s$ (see (17) for the relevant definition), we have:

$$\|f\|_{W_2^s} = \Big( \sum_{|\boldsymbol{\alpha}| \le s} \|\partial^{\boldsymbol{\alpha}} f(\mathbf{x})\|_{L^2}^2 \Big)^{1/2}$$

$$= \Big( \sum_{|\boldsymbol{\alpha}| \le s} \Big\| \partial^{\boldsymbol{\alpha}} \sum_{i=1}^{M} \delta_i g_i(\mathbf{x}) \Big\|_{L^2}^2 \Big)^{1/2}$$

$$= \Big( \sum_{|\boldsymbol{\alpha}| \le s} \Big\| \sum_{i=1}^{M} \delta_i \partial^{\boldsymbol{\alpha}} g_i(\mathbf{x}) \Big\|_{L^2}^2 \Big)^{1/2}$$

$$= \Big( \sum_{|\boldsymbol{\alpha}| \le s} \int_D \Big| \sum_{i=1}^{M} \delta_i \partial^{\boldsymbol{\alpha}} g_i(\mathbf{x}) \Big|^2 d\mathbf{x} \Big)^{1/2}$$

$$= \Big( \sum_{|\boldsymbol{\alpha}| \le s} \sum_{i=1}^{M} \int_{D_i} \big| \partial^{\boldsymbol{\alpha}} g_i(\mathbf{x}) \big|^2 d\mathbf{x} \Big)^{1/2} \tag{23}$$

$$= \Big( \sum_{i=1}^{M} \|g_i(\mathbf{x})\|_{W_2^s}^2 \Big)^{1/2}$$

$$= \sqrt{M} \|g_0\|_{W_2^s}, \tag{24}$$

where (23) holds since the regions $D_1, \ldots, D_M$ are disjoint.

For non-integer values of $s = \nu + \frac{d}{2}$, there always exists $n \in \mathbb{N}^+$ such that $n < s < n+1$, and we can proceed analogously to the analysis in (Hesse, 2006). Note also that our assumption $\nu + \frac{d}{2} \ge 1$ ensures that $n \ge 1$. From the definition of fractional Sobolev norm in 19, and using Hölder's inequality, we can "interpolate" the

fractional Sobolev norm between two integer Sobolev norms (we specify the constants $\lambda$, $\eta$, $p$, and $q$ below):

$$\|f\|_{W_2^s}^2 = \inf_{f'(\mathbf{x})=f(\mathbf{x}),\forall \mathbf{x}\in D, f'\in W_2^s(\mathbb{R}^d)}\left\{\int_{\mathbb{R}^d}(1+\|\mathbf{x}\|_2^2)^s|\hat{f}(\boldsymbol{\xi})|^2 d\mathbf{x}\right\}$$

$$= \int_{\mathbf{x}\in D}(1+\|\mathbf{x}\|_2^2)^s|\hat{f}(\boldsymbol{\xi})|^2 d\mathbf{x} \tag{25}$$

$$= \int_{\mathbf{x}\in D}(1+\|\mathbf{x}\|_2^2)^{s-\lambda}|\hat{f}(\boldsymbol{\xi})|^{2-\eta}\cdot(1+\|\mathbf{x}\|_2^2)^{\lambda}|\hat{f}(\boldsymbol{\xi})|^{\eta} d\mathbf{x}$$

$$\leq \left(\int_{\mathbf{x}\in D}(1+\|\mathbf{x}\|_2^2)^{(s-\lambda)p}|\hat{f}(\boldsymbol{\xi})|^{(2-\eta)p} d\mathbf{x}\right)^{\frac{1}{p}}\cdot\left(\int_{\mathbf{x}\in D}(1+\|\mathbf{x}\|_2^2)^{\lambda q}|\hat{f}(\boldsymbol{\xi})|^{\eta q} d\mathbf{x}\right)^{\frac{1}{q}} \tag{26}$$

$$= \|f\|_{W_2^n}^{2n+2-2s}\cdot\|f\|_{W_2^{n+1}}^{2s-2n} \tag{27}$$

$$= M\cdot\|g_0\|_{W_2^n}^{2n+2-2s}\cdot\|g_0\|_{W_2^{n+1}}^{2s-2n} \tag{28}$$

$$= O\left(M\cdot\|g_0\|_{\mathcal{H}_{n-\frac{d}{2}}}^{2n+2-2s}\cdot\|g_0\|_{\mathcal{H}_{n+1-\frac{d}{2}}}^{2s-2n}\right) \tag{29}$$

$$= O\left(M\frac{\epsilon^{2n+2-2s}}{(w^{n-\frac{d}{2}})^{2n+2-2s}}\cdot\frac{\epsilon^{2s-2n}}{(w^{n+1-\frac{d}{2}})^{2s-2n}}\right) \tag{30}$$

$$= O\left(\frac{M\epsilon^2}{w^{2s-d}}\right) \tag{31}$$

$$= O\left(\frac{M\epsilon^2}{w^{2\nu}}\right), \tag{32}$$

where:

- (25) holds since we are considering functions with support only within the subset $D$, and extending them to $\mathbb{R}^d$ would simply set $f'(\mathbf{x})$ to zero outside of $D$.

- In (26)–(27), we have chosen $\lambda = (n+1)(s-n)$, $\eta = 2(s-n) \in (0,2)$, $p = \frac{1}{n+1-s} > 1$ and $q = \frac{1}{s-n} > 0$, so that the pre-condition of Hölder's inequality holds ($\frac{1}{p}+\frac{1}{q}=1$).

- In addition, (27) uses the identities $(s-\lambda)p = n$, $(2-\eta)p = 2$, and $\lambda q = n+1$, which follow from direct substitutions and simplifications.

- (28) directly applies (24).

- (29) uses Lemma 1, which establishes the equivalence of norms between the Sobolev space $W_2^n$ and Matérn RKHS $\mathcal{H}_\nu$ with parameter $\nu = n - \frac{d}{2}$ (i.e., the notation $\mathcal{H}_\nu$ is equivalent to $\mathcal{H}_M$, but shows the explicit dependence on $\nu$).

- (30) substitutes the RKHS norm bound for $g_0$, i.e., $\|g_0\|_{\mathcal{H}_M} = O(\frac{\epsilon}{w^\nu})$, as demonstrated in Lemma 2.

- (31) cancels $(w^{n-\frac{d}{2}})^{2n-2s}$ with $(w^{n-\frac{d}{2}})^{2s-2n}$, so that the exponent to $w$ becomes $(2n-d)+(2s-2n) = 2d-s$.

- (32) uses $s = \nu + \frac{d}{2}$.

Again using the norm equivalence between Sobolev function and the Matérn RKHS, we can bound the RKHS norm of $f$ as follows in view of (32) and (21):

$$\|f\|_{\mathcal{H}_M} = O\left(\frac{\epsilon}{M^{\frac{\nu}{d}+\frac{1}{2}}}\right). \tag{33}$$

Equating the right-hand side of (33) with $B$ and rearranging, it follows that $\|f\|_{\mathcal{H}_M} \leq B$ with a choice of $\epsilon$ satisfying

$$\epsilon = \Theta\left(\frac{B}{M^{\frac{\nu}{d}+\frac{1}{2}}}\right). \tag{34}$$

## C.2 Completion of the Proof of Theorem 3

The finite function (sub-)class described in Section C.1 is denoted by $\overline{\mathcal{H}}_{\mathrm{M}} \subset \mathcal{H}_{\mathrm{M}}$. We first consider the worst-case criterion. Suppose there is an algorithm estimating $I$ for a function $f \in \overline{\mathcal{H}}_{\mathrm{M}}$, and the algorithm has a budget of $T = \frac{M}{2}$. The best this algorithm can do is determine the signs of $\frac{M}{2}$ out of the $M$ bumps. For the unexplored regions, being in the worst-case setting, we can make an adversarial argument: The adversary either makes all of these bumps negative or all of them positive, whichever leads to a higher error. This leads to two feasible values of $I$ that differ by $\Theta(MI_0)$, which in turn implies that adversarially choosing the worse of the two must give $|I - \hat{I}| = \Omega(MI_0)$. Hence, when $T = \frac{M}{2}$, we have

$$\mathcal{E}_{\mathrm{wst}}^{\mathrm{M}}(T) \geq \Omega\left(\frac{M}{2} \cdot I_0\right) \tag{35}$$

$$= \Omega(Mw^d\epsilon) \tag{36}$$

$$= \Omega(\epsilon), \tag{37}$$

where (36) uses (22), and (37) uses (21). Substituting (34) and $T = \frac{M}{2}$ into (37), we obtain

$$\mathcal{E}_{\mathrm{wst}}^{\mathrm{M}}(T) = \Omega\left(\frac{B}{(2T)^{\frac{\nu}{d}+\frac{1}{2}}}\right) = \Omega\left(\frac{1}{T^{\frac{\nu}{d}+\frac{1}{2}}}\right),$$

which establishes the desired worst-case lower bound.

For the average-case criterion, we proceed slightly differently. We first note that the supremum in $\mathcal{E}_{\mathrm{avg}}^{\mathrm{M}}(T) = \sup_{f \in \mathcal{H}_{\mathrm{M}}} \mathbb{E}\big[|I - \hat{I}|\big]$ can be lower bounded by the average with respect to $f$ drawn uniformly from $\overline{\mathcal{H}}_{\mathrm{M}}$ (and also still averaged over any randomness in the algorithm). Again considering a budget of $T = \frac{M}{2}$, there must be at least $\frac{M}{2}$ regions with no samples, and for each such region, the associated integral is $+I_0$ or $-I_0$ with equal probability. Thus, conditioned on the observed samples, the posterior distribution of $I$ can be expressed as a sum of two independent random variables, one of which is of the form

$$\mathcal{S}_{\frac{M}{2}} = \sum_{i=1}^{M/2} \delta_i \cdot I_0, \tag{38}$$

where for notational convenience, we assume (without loss of generality) that it is the first $\frac{M}{2}$ regions that have no samples. That is, $\mathcal{S}_{\frac{M}{2}}$ is a random variable expressing the posterior uncertainty in the $\frac{M}{2}$ non-sampled regions. There may also be additional uncertainty due *more than $\frac{M}{2}$ regions* being non-sampled, but for proving a lower bound, it suffices to consider the case that there is no additional uncertainty beyond $\mathcal{S}_{\frac{M}{2}}$.

Since $\delta_i$ is equiprobable on $\{+1, -1\}$, we have $\mathbb{E}[\mathcal{S}_{\frac{M}{2}}] = 0$. Moreover, the variance of (38) is given by

$$\sigma_{\mathcal{S}_{\frac{M}{2}}}^2 = \sum_{i=1}^{M/2} \mathrm{Var}(\delta_i \cdot I_0) = \frac{M}{2}I_0^2, \tag{39}$$

since $\delta_1 I_0, \cdots, \delta_{\frac{M}{2}} I_0$ are i.i.d. random variables each having variance $I_0^2$. In the limit of large $M$, if we consider the normalized sum

$$Z_{\mathcal{S}_{\frac{M}{2}}} = \frac{\mathcal{S}_{\frac{M}{2}} - \mathbb{E}[\mathcal{S}_{\frac{M}{2}}]}{\sigma_{\mathcal{S}_{\frac{M}{2}}}},$$

then by the *central limit theorem* (CLT) (Feller, 1957, Ch. VIII), as $M \to \infty$, $Z_{\mathcal{S}_{\frac{M}{2}}}$ convergences in distribution to the standard normal distribution $\mathcal{N}(0,1)$. In other words $\mathcal{S}_{\frac{M}{2}}$ is asymptotically distributed as $\mathcal{N}(0, \sigma_{\mathcal{S}_{\frac{M}{2}}}^2)$. When estimating a Gaussian (or an asymptotically Gaussian quantity), an average absolute error proportional to the standard deviation is unavoidable; for instance, if we know that $Z \sim \mathcal{N}(0, \sigma_Z^2)$,

then our best guess of $Z$ is $\hat{Z} = 0$, but this still gives $\mathbb{E}[|Z - \hat{Z}|] = \mathbb{E}[|Z|] = \sigma\sqrt{\frac{2}{\pi}}$. Thus, the lower bound for the average case regret is $\mathcal{E}^{\mathrm{M}}_{\mathrm{avg}}(T) \geq \Omega(\sigma_{\mathcal{S}_{\frac{M}{2}}}) = \Omega(\sqrt{M}I_0)$, and by substituting (21), (22), (34), we obtain

$$\mathcal{E}^{\mathrm{M}}_{\mathrm{avg}}(T) = \Omega\Big(\frac{\epsilon}{\sqrt{M}}\Big) = \Omega\Big(\frac{1}{T^{\frac{\nu}{d}+1}}\Big).$$

## D Unified Derivation of Both Terms in Theorem 1 (Lower Bound)

In this appendix, we show that analyzing a suitably chosen hard subset of functions provides an alternative derivation of both terms in Theorem 1 in a unified manner. Specifically, we consider the same class $\overline{\mathcal{H}}_{\mathrm{M}}$ that was used in the noiseless case in Appendix C, but the analysis requires substantial modifications and is much more technical. We proceed in several steps.

**Step 1: Reduction to a simpler problem.** Similarly to the noiseless setting above, we start by lower bounding $\mathcal{E}^{\mathrm{M}}_{\mathrm{avg}}(T, \sigma) = \sup_{f \in \mathcal{H}_{\mathrm{M}}} \mathbb{E}[|I - \hat{I}|]$ by $\mathbb{E}[|I - \hat{I}|]$, where now the average is taken over three sources of randomness: the uniform distribution over $2^M$ functions in $\overline{\mathcal{H}}_{\mathrm{M}}$ (with parameters $M$ and $\epsilon$), the randomization in the algorithm, and the noise. Moreover, once $f$ is randomized, letting the algorithm be deterministic is without loss of optimality, so we assume this is the case (this is simply an instantiation of Yao's minimax principle).

We claim that in order to attain a lower bound with the preceding prior, it suffices to attain a lower bound in the following simplified setup:

- There exists an unknown collection of signs $S_i \in \{-1, +1\}$ for $i = 1, \dots, M$, each taking either value independently with probability $\frac{1}{2}$.

- The goal of the algorithm is to estimate $I_0 \sum_{i=1}^{M} S_i$, where $I_0$ is the integral of a single (positive) bump in the original BQ problem.

- At time $t$, the algorithm may select an index $i_t$ (possibly in an adaptive manner) and observe $y_t = \epsilon S_{i_t} + \epsilon_t$ with independent noise $\epsilon_t \sim N(0, \sigma^2)$.

We observe that this problem is exactly equivalent to our original problem in the case that the BQ algorithm is constrained to select midpoints of the bumps, where the bump takes its highest absolute value (i.e., $\epsilon$). Intuitively, this is without loss of optimality because such points have the highest signal, and are thus the most informative.

To make this more formal, we note that sampling a point with absolute value $|f(x)| = c \in (0, \epsilon)$ gives $y_t = cS_{i_t} + \epsilon_t$ with $S_{i_t}$ being the associated bump sign, and this is information-theoretically equivalent to observing $y_t \frac{\epsilon}{c} = \epsilon S_{i_t} + \frac{\epsilon}{c}\epsilon_t$. Since $\frac{\epsilon}{c} > 1$, this simply amounts to still observing $\epsilon S_{i_t}$, but with more noise, and this extra noise could always be artificially added anyway. Hence, sampling at the midpoint is without loss of generality or optimality.[7]

We proceed by studying this simplified problem.

**Step 2: Establish hardness of estimating most sign values.** As a stepping stone to characterizing the difficulty of estimating $I_0 \sum_{i=1}^{M} S_i$, we provide an auxiliary result on the hardness of estimating $\mathbf{S} = (S_1, \dots, S_M)$ to within a certain Hamming distance. Although estimating each individual sign is a harder problem than estimating their sum (which may appear concerning from the perspective of proving a lower bound), this will turn out to be a useful intermediate step. We let $\widehat{\mathbf{S}} = (\widehat{S}_1, \dots, \widehat{S}_M)$ denote an estimate of $\mathbf{S}$ based on the queries.

**Lemma 3.** *In the simplified setup with discrete queries $i_t \in \{1, \dots, M\}$ (rather than $x_t \in D$), consider any (possibly adaptive) deterministic algorithm that produces an estimate $\widehat{\mathbf{S}}$ of $\mathbf{S}$. Then, there exists a sufficiently*

---

[7]The locations with $|f(x)| = 0$ carry no information, so we can assume without loss of generality that they are never sampled.

*small constant c such that we require a time horizon of*

$$T \geq c \cdot M \cdot \max\left\{1, \frac{\sigma^2}{\epsilon^2}\right\} \tag{40}$$

*in order to obtain* $\mathbb{E}\big[d_{\mathrm{H}}(\mathbf{S}, \widehat{\mathbf{S}})\big] \leq \frac{M}{8}$. *Here* $d_{\mathrm{H}}$ *denotes the Hamming distance, and the expectation is with respect to* $\mathbf{S}$ *uniform on the* $2^M$ *possibilities, as well as the random noise.*

**Proof.** By a standard variant of Fano's inequality with approximate recovery (e.g., see (Scarlett & Cevher, 2019, Thm. 2)), we have

$$\mathbb{P}\left[d_{\mathrm{H}}(\mathbf{S}, \widehat{\mathbf{S}}) \geq \frac{M}{4}\right] \geq 1 - \frac{I(\mathbf{S}; \widehat{\mathbf{S}}) + 1}{\log(2^M) - \log(N_{\max})}, \tag{41}$$

where $I(\cdot; \cdot)$ denotes the mutual information (Cover & Thomas, 2006), and $N_{\max}$ is the number of vectors in $\{-1, 1\}^M$ within Hamming distance $\frac{M}{4}$ of any fixed vector (e.g., the all-ones vector). We have $N_{\max} = \sum_{i=0}^{M/4} \binom{M}{i} \leq M\binom{M}{M/4}$, from which a standard bound on the binomial coefficient gives $N_{\max} \leq e^{MH_2(1/4)(1+o(1))}$ with $H_2(q) = q\log\frac{1}{q} + (1-q)\log\frac{1}{1-q}$ being the binary entropy function. Since $H_2(1/4)$ is strictly smaller than $\log 2$, substitution into (41) gives

$$\mathbb{P}\left[d_{\mathrm{H}}(\mathbf{S}, \widehat{\mathbf{S}}) \geq \frac{M}{4}\right] \geq 1 - \frac{I(\mathbf{S}; \widehat{\mathbf{S}}) + 1}{\Theta(M)}. \tag{42}$$

Moreover, following standard steps, we can upper bound the numerator as follows (Scarlett & Cevher, 2019, Sec. 3):

- Use the data processing inequality to write $I(\mathbf{S}; \widehat{\mathbf{S}}) \leq I(\mathbf{S}; \mathbf{I}, \mathbf{Y})$, with $(\mathbf{I}, \mathbf{Y})$ being the length-$T$ collection of sampled inputs and observed outputs by the algorithm.

- Use the chain rule for mutual information to upper bound $I(\mathbf{S}; \mathbf{I}, \mathbf{Y})$ by a corresponding sum over time indices: $I(\mathbf{S}; \mathbf{I}, \mathbf{Y}) \leq \sum_{t=1}^{T} I(\mathbf{S}; Y_t | I_t)$.

To simplify the last expression, we note that given $I_t$, the only entry of $\mathbf{S}$ that impacts $Y_t$ is $S_{I_t}$, so we can further write $I(\mathbf{S}; Y_t | I_t) \leq I(S_{I_t}; Y_t | I_t)$.

Recall that when $S_{I_t}$ equals some value $s \in \{-1, 1\}$, the corresponding observation is $y_t \sim N(sI_0, \sigma^2)$. Hence, by the relation between mutual information and KL divergence (Scarlett & Cevher, 2019, Sec. 3.3), the preceding mutual information is further upper bounded by the KL divergence between $N(\pm\epsilon, \sigma^2)$ and $N(0, \sigma^2)$ (this is the same regardless of whether $\epsilon$ has a $+1$ or $-1$ coefficient), which is $\frac{\epsilon^2}{2\sigma^2}$.

Substituting the preceding findings back into the preceding inequality $I(\mathbf{S}; \mathbf{I}, \mathbf{Y}) \leq \sum_{t=1}^{T} I(\mathbf{S}; Y_t | I_t)$, it follows that $I(\mathbf{S}; \mathbf{I}, \mathbf{Y}) \leq \frac{T\epsilon^2}{2\sigma^2}$. Hence, if $T < c \cdot \frac{M\sigma^2}{\epsilon^2}$ with a small enough constant $c$, then the right-hand side of (42) exceeds $\frac{1}{2}$. The fact that $d_{\mathrm{H}}(\mathbf{S}, \widehat{\mathbf{S}}) \geq \frac{M}{4}$ with probability exceeding $\frac{1}{2}$ then implies that $\mathbb{E}\big[d_{\mathrm{H}}(\mathbf{S}, \widehat{\mathbf{S}})\big] > \frac{M}{8}$.

The preceding argument proves the lemma when $\frac{\sigma^2}{\epsilon^2} \geq 1$. On the other hand, if $\frac{\sigma^2}{\epsilon^2} < 1$, then the requirement in (40) simply becomes $T \geq cM$, which we claim to be trivially necessary for attaining $\mathbb{E}\big[d_{\mathrm{H}}(\mathbf{S}, \widehat{\mathbf{S}})\big] \leq \frac{M}{8}$, as long as $c \leq \frac{3}{4}$. To see this, note that with any smaller number of samples, a quarter (or more) of the indices cannot even be sampled once. When this is the case, the algorithm cannot do any better than guessing the corresponding $S_i$ values, getting each one correct with probability $\frac{1}{2}$. $\qquad\square$

The contrapositive statement of Lemma 3 is that if $T < c \cdot M \cdot \max\left\{1, \frac{\sigma^2}{\epsilon^2}\right\}$, then it must hold that $\mathbb{E}\big[d_{\mathrm{H}}(\mathbf{S}, \widehat{\mathbf{S}})\big] > \frac{M}{8}$. Furthermore, by writing the Hamming distance as a sum of indicator function $\mathbb{1}\{S_i \neq \widehat{S}_i\}$, the preceding inequality can be written as

$$\frac{1}{M}\sum_{i=1}^{M} \mathbb{P}[S_i \neq \widehat{S}_i] > \frac{1}{8}. \tag{43}$$

**Step 3: Characterize the posterior uncertainty.** In the argument that follows, we are not directly interested in $\mathbb{P}[S_i \neq \widehat{S}_i]$, but instead $\mathbb{P}[S_i \neq \widehat{S}_i \,|\, \mathcal{D}]$, where $\mathcal{D} = (\mathbf{I}, \mathbf{Y})$ contains the $T$ pairs of the form $(i_t, y_t)$ collected throughout the course of the algorithm. This conditional probability can be viewed as representing the posterior uncertainty of $S_i$, with a value of $\frac{1}{2}$ meaning complete uncertainty, and a value of 0 or 1 meaning complete certainty.

The probability in (43) is taken with respect to the joint randomness of $\mathbf{S}$ and the noise (which enters via $\mathcal{D}$). While the decomposition $\mathbb{P}[\mathbf{S}]\mathbb{P}[\mathcal{D}|\mathbf{S}]$ is most natural, it is useful to consider the opposite form $\mathbb{P}[\mathcal{D}]\mathbb{P}[\mathbf{S}|\mathcal{D}]$, so that we can analyze the *posterior distribution* $\mathbb{P}[\mathbf{S}|\mathcal{D}]$.

With this in mind, we claim that the following holds with probability at least $\frac{1}{16}$ *with respect to* $\mathcal{D}$ (i.e., the equation to follow depends on $\mathcal{D}$ but still contains randomness via $\mathbb{P}[\mathbf{S}|\mathcal{D}]$):

$$\frac{1}{M} \sum_{i=1}^{M} \mathbb{P}[S_i \neq \widehat{S}_i \,|\, \mathcal{D}] > \frac{1}{16}. \tag{44}$$

To see this, assume by contradiction that this were only to hold with probability less than $\frac{1}{16}$. Then, letting $\mathcal{A}$ denote the event that (44) holds, we would have

$$\frac{1}{M} \sum_{i=1}^{M} \mathbb{P}[S_i \neq \widehat{S}_i] = \mathbb{E}\left[ \frac{1}{M} \sum_{i=1}^{M} \mathbb{P}[S_i \neq \widehat{S}_i | \mathcal{D}] \right]$$

$$= \mathbb{E}\left[ \frac{1}{M} \sum_{i=1}^{M} \mathbb{P}[S_i \neq \widehat{S}_i | \mathcal{D}] \, \mathbb{1}\{\mathcal{D} \in \mathcal{A}\} \right] + \mathbb{E}\left[ \frac{1}{M} \sum_{i=1}^{M} \mathbb{P}[S_i \neq \widehat{S}_i | \mathcal{D}] \, \mathbb{1}\{\mathcal{D} \notin \mathcal{A}\} \right].$$

where the first line uses the tower property of expectation. Then, the two terms are bounded as follows:

- Using what we assumed by contradiction and upper bounding $\frac{1}{M} \sum_{i=1}^{M} \mathbb{P}[S_i \neq \widehat{S}_i \,|\, \mathcal{D}] \leq 1$, the first term is at most $\frac{1}{16}$.

- Using the opposite inequality to (44) for $\mathcal{D} \notin \mathcal{A}$, and upper bounding $\mathbb{1}\{\cdot\} \leq 1$, the second term is also at most $\frac{1}{16}$.

Thus, we obtain $\frac{1}{M} \sum_{i=1}^{M} \mathbb{P}[S_i \neq \widehat{S}_i] \leq \frac{1}{8}$, which contradicts (43), and we conclude that (44) must hold with probability at least $\frac{1}{16}$ (with respect to $\mathcal{D}$).

We now apply a similar argument to the preceding one, but considering the uniform distribution over $M$ implicit in (44) (as opposed to the distribution of $\mathcal{D}$). Omitting the details to avoid repetition, it follows that at least $\frac{M}{32}$ of the indices in $\{1, \ldots, M\}$ have $\mathbb{P}[S_i \neq \widehat{S}_i | \mathcal{D}] > \frac{1}{32}$; any smaller number than $\frac{M}{32}$ would contradict (44).

The above findings are summarized in the following lemma.

**Lemma 4.** *Under the setup of Lemma 3, if*

$$T < c \cdot M \cdot \max\left\{ 1, \frac{\sigma^2}{\epsilon^2} \right\} \tag{45}$$

*for sufficiently small $c > 0$, then with probability at least $\frac{1}{16}$ (with respect to $\mathcal{D}$), there exist at least $\frac{M}{32}$ indices such that $\mathbb{P}[S_i \neq \widehat{S}_i \,|\, \mathcal{D}] > \frac{1}{32}$.*

**Step 4: Central limit theorem.** We now return to the problem formed in Step 1, where the goal is to estimate $I_0 \sum_{i=1}^{M} S_i$, and the algorithm does not necessarily form any entry-by-entry estimate $\widetilde{\mathbf{S}}$. We continue to let $\mathcal{D}$ denote the samples collected, and we let $\mathcal{D}_i$ denote the subset of $\mathcal{D}$ corresponding to times when $i_t = i$.

**Lemma 5.** *Under the uniform prior on $\mathbf{S} = (S_1, \ldots, S_M)$, conditioned on any collection of samples $\mathcal{D}$, we have that the signs $(S_1, \ldots, S_M)$ remain conditionally independent.*

**Proof.** This is immediate from the fact that we consider an independent prior (namely, the uniform prior over all $2^M$ sign patterns) and assume that the noise terms between times are independent. Thus, whenever some index $i_t$ is selected, the resulting observation $y_t$ bears information about $S_{i_t}$, but bears no information about any of the other $S_j$. $\qquad\square$

By Lemma 5, conditioned on $\mathcal{D}$, the posterior distribution of $\sum_{i=1}^{M} S_i$ is a sum of independent $\pm 1$-valued random variables. Moreover, by Lemma 4, when $T$ satisfies (45) it holds with probability at least $\frac{1}{16}$ that at least $\frac{1}{32}$ fraction of these indices have strictly positive posterior variance. This, in turn, implies that $I_0 \sum_{i=1}^{M} S_i$ has a posterior variance of $\Omega(M I_0^2)$.

Having a constant fraction of strictly positive-variance terms is sufficient for applying the central limit theorem for independent but non-identical random variables (Feller, 1957, Ch. VIII). By doing so, we find that $I_0 \sum_{i=1}^{M} S_i$ is asymptotically Gaussian; the mean is inconsequential for our purposes, and the variance scales as $\Omega(M I_0^2)$, i.e., the standard deviation is $\Omega(\sqrt{M} \cdot I_0)$. As highlighted in Appendix C, when we have a posterior standard deviation of $\Omega(\sqrt{M} \cdot I_0)$, we incur $\Omega(\sqrt{M} \cdot I_0)$ error. Since we have shown that this is the case with constant probability, it follows that the average error is $\Omega(\sqrt{M} \cdot I_0)$.

**Step 5: Simplification.** Recall from Appendix C that in our function class, we have $M = \lfloor \frac{1}{w} \rfloor^d$, $I_0 = \Theta(w^d \epsilon) = \Theta(\frac{\epsilon}{M})$, and $\epsilon = \Theta(\frac{1}{M^{\nu/d+1/2}})$. Hence, the scaling $\Omega(\sqrt{M} \cdot I_0)$ can be expressed as $\Omega(\frac{\epsilon}{\sqrt{M}})$. We now complete the proof by considering two cases:

- If the maximum in (45) is achieved by the first term, then we have $M = \Theta(T)$ (supposing that $T$ is as high as possible subject to (45)). Moreover, the above-established fact gives an $\Omega(\frac{1}{T^{\frac{\nu}{d}+1}})$ lower bound.

- If the maximum in (45) is achieved by the second term, then we get $M = \Theta(\frac{T\epsilon^2}{\sigma^2})$, or equivalently $\frac{\epsilon}{\sqrt{M}} = \Theta(\frac{\sigma}{\sqrt{T}})$. Hence, the lower bound is $\Omega(\frac{\sigma}{\sqrt{T}})$ for both kernels.

Combining these two cases, we obtain a final lower bound of $\mathcal{E}_{\mathrm{avg}}^{\mathrm{M}}(T, \sigma) = \Omega\big(\max\big\{\frac{\sigma}{\sqrt{T}}, \frac{1}{T^{\frac{\nu}{d}+1}}\big\}\big)$. This is equivalent to $\Omega(T^{\frac{\nu}{d}-1} + \sigma T^{-\frac{1}{2}})$, and the proof of Theorem 1 is complete.

# E Proofs of Average-Case Upper Bounds

## E.1 Proof of Theorem 2 (General Guarantee for Algorithm 1)

Our first result establishes that $\hat{R}$ is an unbiased estimator of $R$.

**Lemma 6.** *Condition on arbitrary fixed values of $y_1, \ldots, y_{T/2}$ (and hence, fixed $\hat{I}_1$ and $R$), and consider the resulting distribution of $\hat{R}$ due to the randomness in $\mathbf{x}_{T/2+1}, \ldots, \mathbf{x}_T$ and $\epsilon_{T/2+1}, \ldots, \epsilon_T$. We have*

$$\mathbb{E}[\hat{R}] = R. \tag{46}$$

**Proof.** Recalling that $\hat{f}$ is the initial estimate of $f$ based on the first $\frac{T}{2}$ samples, we have

$$
\begin{aligned}
\mathbb{E}[\hat{R}] &= \mathbb{E}\Big[\frac{2}{T}\sum_{t=T/2+1}^{T}\big(y_t - \hat{f}(\mathbf{x}_t)\big)\Big] \\
&= \frac{2}{T}\sum_{t=T/2+1}^{T}\mathbb{E}[y_t - \hat{f}(\mathbf{x}_t)] \\
&= \frac{2}{T}\sum_{t=T/2+1}^{T}\mathbb{E}[f(\mathbf{x}_t) - \hat{f}(\mathbf{x}_t) + \epsilon_t] \\
&= \frac{2}{T}\sum_{t=T/2+1}^{T}\mathbb{E}[f(\mathbf{x}_t) - \hat{f}(\mathbf{x}_t)] + \frac{2}{T}\sum_{t=T/2+1}^{T}\mathbb{E}[\epsilon_t] \\
&= \frac{2}{T}\sum_{t=T/2+1}^{T}\int_D p(\mathbf{x})[f(\mathbf{x}) - \hat{f}(\mathbf{x})]d\mathbf{x} + 0 \\
&= \int_D p(\mathbf{x})[f(\mathbf{x}) - \hat{f}(\mathbf{x})]d\mathbf{x} = R,
\end{aligned}
$$

where the second and fourth equalities are due to the linearity of expectation, and the fifth equality holds since $\mathbf{x}_t \sim p(\mathbf{x})$ and $\mathbb{E}[\epsilon_t] = 0$. $\qquad\square$

In addition, the variance of the residual estimator $\hat{R}$ is bounded according to the following.

**Lemma 7.** *Under the setup of Lemma 6, we have*

$$
\mathrm{Var}[\hat{R}] \leq \frac{2p_{\max}}{T}\|f - \hat{f}\|_{L^2}^2 + \frac{2\sigma^2}{T}, \tag{47}
$$

*where the variance is with respect to the randomness in $\mathbf{x}_{T/2+1}, \ldots, \mathbf{x}_T$ and $\epsilon_{T/2+1}, \ldots, \epsilon_T$.*

**Proof.** We have

$$
\begin{aligned}
\mathrm{Var}[\hat{R}] &= \mathrm{Var}\Big[\frac{2}{T}\sum_{t=T/2+1}^{T}\big(y_t - \hat{f}(\mathbf{x}_t)\big)\Big] \\
&= \frac{4}{T^2}\mathrm{Var}\Big[\sum_{t=T/2+1}^{T}\big(f(\mathbf{x}_t) - \hat{f}(\mathbf{x}_t) + \epsilon_t\big)\Big] \\
&= \frac{4}{T^2}\sum_{t=T/2+1}^{T}\mathrm{Var}\Big[f(\mathbf{x}_t) - \hat{f}(\mathbf{x}_t) + \epsilon_t\Big] \\
&= \frac{4}{T^2}\sum_{t=T/2+1}^{T}\mathrm{Var}[f(\mathbf{x}_t) - \hat{f}(\mathbf{x}_t)] + \frac{4}{T^2}\cdot\sum_{t=T/2+1}^{T}\mathrm{Var}[\epsilon_t] \\
&= \frac{4}{T^2}\sum_{t=T/2+1}^{T}\mathrm{Var}[f(\mathbf{x}_t) - \hat{f}(\mathbf{x}_t)] + \frac{2\sigma^2}{T}, \tag{48}
\end{aligned}
$$

where on the third line we use the independence of $\mathbf{x}_t$ and $\epsilon_t$ across $t$, on the fourth line we use the fact that $\epsilon_t$ is independent of $f(\mathbf{x}_t) - \hat{f}(\mathbf{x}_t)$, and the last line is due to $\mathrm{Var}[\epsilon_t] = \sigma^2$. Moreover, by the definition of

variance, we have

$$
\begin{aligned}
\mathrm{Var}[f(\mathbf{x}_t) - \hat{f}(\mathbf{x}_t)] &= \mathbb{E}[(f(\mathbf{x}_t) - \hat{f}(\mathbf{x}_t))^2] - \mathbb{E}[f(\mathbf{x}_t) - \hat{f}(\mathbf{x}_t)]^2 \\
&\leq \mathbb{E}[(f(\mathbf{x}_t) - \hat{f}(\mathbf{x}_t))^2] \\
&= \int_D p(\mathbf{x})(f(\mathbf{x}) - \hat{f}(\mathbf{x}))^2 d\mathbf{x} \\
&\leq p_{\max} \|f - \hat{f}\|_{L^2}^2,
\end{aligned}
\tag{49}
$$

where the last inequality is due to our assumption that $p(\mathbf{x}) \in [0, p_{\max}]$ for all $\mathbf{x}$. Substituting (49) into (48), we obtain (47) as desired. $\qquad\square$

We can now analyze the error of our algorithm averaged over *all* samples, including the first $T/2$. Let $\mathbb{E}_1[\cdot]$ (respectively, $\mathbb{E}_2[\cdot]$) denote averaging with respect to the randomness from the first (respectively, second) batch. We first note that conditioned on the first $T/2$ samples, we have

$$
\begin{aligned}
\mathbb{E}_2\Big[\big|I - \hat{I}_1 - \hat{R}\big|\Big] &\leq \sqrt{\mathbb{E}_2\Big[\big(I - \hat{I}_1 - \hat{R}\big)^2\Big]} \\
&= \sqrt{\mathbb{E}_2\Big[\big(R - \hat{R}\big)^2\Big]} \\
&= \sqrt{\mathbb{E}_2\Big[\big(\mathbb{E}_2[\hat{R}] - \hat{R}\big)^2\Big]} \\
&= \sqrt{\mathrm{Var}[\hat{R}]} \\
&\leq \sqrt{2p_{\max}}T^{-\frac{1}{2}}\|f - \hat{f}\|_{L^2} + \sqrt{2}\sigma T^{-\frac{1}{2}},
\end{aligned}
\tag{50}
\tag{51}
$$

where the first inequality follows from Jensen's inequality, the third line holds due to Lemma 6, and the last two steps are due to Lemma 7 and the elementary inequality $\sqrt{x+y} \leq \sqrt{x} + \sqrt{y}$.

Then, incorporating the randomness from the first $T/2$ samples, we obtain

$$
\begin{aligned}
\mathbb{E}\Big[\big|I - \hat{I}_1 - \hat{R}\big|\Big] &= \mathbb{E}_1\Big[\mathbb{E}_2\Big[\big|I - \hat{I}_1 - \hat{R}\big|\Big]\Big] \\
&\leq \sqrt{2p_{\max}}T^{-\frac{1}{2}}\mathbb{E}\big[\|f - \hat{f}\|_{L^2}\big] + \sqrt{2}\sigma T^{-\frac{1}{2}},
\end{aligned}
\tag{52}
$$

where we applied the tower property of expectation, followed by (51) (note that $\mathbb{E}_1\big[\|f - \hat{f}\|_{L^2}^2\big] = \mathbb{E}\big[\|f - \hat{f}\|_{L^2}\big]$ since no quantities from the second batch are present). This concludes the proof of Theorem 2.

### E.2 Proofs of Corollaries 1 and 2 (Noisy Matérn Upper Bounds)

It remains to upper bound the average-case $L^2$-error $\mathbb{E}\big[\|f - \hat{f}\|_{L^2}\big]$ between the true function $f$ and the estimate $\hat{f}$. To do so, we will use results from (Wynne et al., 2021) on the $L^2$ estimation error for functions in Sobolev spaces, and adapt them to our problem for functions in the Matérn RKHS. These results depend on various technical assumptions made in (Wynne et al., 2021), some of which are trivially satisfied in our setting: our domain $[0,1]^d$ implies their Assumption 1, our Matérn-$\nu$ functions with finite RKHS norm implies their Assumptions 2, 4 and 5. Moreover, their Assumption 3 only pertains to the misspecified setting, imposing the requirement that $\{\hat{\nu}_t\}_{t \geq T}$ has finitely many values (as we also assume for Corollary 2).

We consider the commonly-used notions of *fill distance* $h_X$ and *separation radius* $q_X$, which are widely used (e.g., see (Wynne et al., 2021; Rieger & Zwicknagl, 2010)) and the associated convergence rates for kernel-based interpolation methods. For a point set $X \subset D$, the two distances are defined as

$$
h_X = \sup_{\mathbf{x} \in X} \inf_{\mathbf{y} \in D} \|\mathbf{x} - \mathbf{y}\|, \quad q_X := \frac{1}{2} \min_{\mathbf{x}, \mathbf{y} \in X, \mathbf{x} \neq \mathbf{y}} \|\mathbf{x} - \mathbf{y}\|.
\tag{53}
$$

Intuitively, sufficiently small fill distance implies that $X$ covers the whole domain $D$. It is known that *quasi-uniform* point sets achieve the optimal order of $h_X$ and $q_X$, at $\Theta(T^{-\frac{1}{d}})$ (see, e.g. (Novak & Woźniakowski,

2008, Thm. 4.17)). Moreover, (Santin & Haasdonk, 2017, Cor. 11) shows that greedily minimizing the GP posterior variance with $\lambda_\sigma = 0$ leads to asymptotically uniform points, which in turn achieves a small fill distance. Considering maximum-variance sampling (Algorithm 2) in the first batch in our meta-algorithm (Algorithm 1), we are able to directly make use of the following results from (Wynne et al., 2021).

**Lemma 8.** ($L^2$-Error Result from (Wynne et al., 2021, Thm. 4)) *Let $f$ be any function in $\mathcal{H}_M$, and $X = \{\mathbf{x}_t\}_{t=1}^T$ be the sequence of points selected by Algorithm 2 with $\lambda_\sigma = 0$, which outputs $\mu_T$ with parameter $\lambda_\mu$. Then, there exists a constant $h_0 > 0$ such that $\forall X \subseteq D$ with $h_X \leq h_0$, and when $\nu$ is known, the average noisy $L^2$-error is*

$$\mathbb{E}[\|f - \mu_T\|_{L^2}] = O\Big( h_X^{\frac{d}{2}} (h_X^\nu + \lambda_\mu) \|f\|_{\mathcal{H}_M} + h_X^{\frac{d}{2}} (h_X^\nu \lambda_\mu^{-1} + 1) \mathbb{E}[\|\boldsymbol{\epsilon}\|] \Big). \tag{54}$$

*where $\boldsymbol{\epsilon} = (\epsilon_1, \ldots, \epsilon_T)$ is the vector of noise terms*

*Moreover, under the modified version of Algorithm 2 for the misspecified setting (described just above Corollary 2, where $\nu^-$ and $\nu^+$ are also defined), we have*

$$\mathbb{E}[\|f - \mu_T\|_{L^2}] = O\Big( h_X^{\frac{d}{2}} \big( h_X^{\min(\nu,\nu^-)+\nu^+-\nu} q_X^{\nu-\nu^+} + \lambda_\mu q_X^{\nu-\nu^+} \big) \|f\|_{\mathcal{H}_M} + h_X^{\frac{d}{2}} \big( h_X^{\nu^-} \lambda_\mu^{-1} + 1 \big) \mathbb{E}[\|\boldsymbol{\epsilon}\|] \Big). \tag{55}$$

To obtain this lemma from (Wynne et al., 2021, Thm. 4), we substitute $s \to 0$, $\tau_f \to \nu + \frac{d}{2}$, $\tau_k^+ \to \nu^+ + \frac{d}{2}$, $\tau_k^- \to \nu^- + \frac{d}{2}$, $q \to 2$ and $m(\cdot) \to 0$ in the notation therein. Setting $\lambda_\mu = \Theta(T^{-\frac{\nu}{d}})$ and $h_X = \Theta(T^{-\frac{1}{d}})$, (54) can be simplified as

$$\mathbb{E}[\|f - \mu_T\|_{L^2}] = O\Big( T^{-\frac{\nu}{d}-\frac{1}{2}} B + T^{-\frac{1}{2}} \mathbb{E}[\|\boldsymbol{\epsilon}\|] \Big). \tag{56}$$

Similarly, by setting $\lambda_\mu = \Theta(T^{-\frac{\nu^-}{d}})$, $h_X = \Theta(T^{-\frac{1}{d}})$, and $q_X = \Theta(T^{-\frac{1}{d}})$, (55) reduces to

$$\mathbb{E}[\|f - \mu_T\|_{L^2}] = O\Big( T^{-\frac{\min(\nu^-,\nu)+\nu^+-\nu-\nu^++\nu}{d}-\frac{1}{2}} B + T^{-\frac{\nu+\nu^--\nu^+}{d}-\frac{1}{2}} B + T^{-\frac{1}{2}} \mathbb{E}[\|\boldsymbol{\epsilon}\|] \Big). \tag{57}$$

Note that since the noises are i.i.d Gaussian with zero mean, we have

$$\mathbb{E}[\|\boldsymbol{\epsilon}\|] \leq \sqrt{\mathbb{E}[\|\boldsymbol{\epsilon}\|^2]} = \sqrt{\mathbb{E}[\epsilon_1^2] + \cdots + \mathbb{E}[\epsilon_{T/2}^2]} = \sqrt{\text{Var}[\epsilon_1] + \cdots + \text{Var}[\epsilon_{T/2}]} = \sqrt{\frac{T}{2}} \sigma. \tag{58}$$

For $f \in \mathcal{H}_M$ and known $\nu$, substituting (58) and (56) into (52), we obtain

$$\mathbb{E}\Big[|I - \hat{I}_1 - \hat{R}|\Big] = O\Big( \sqrt{p_{\max}} T^{-\frac{1}{2}} \big( T^{-\frac{\nu}{d}-\frac{1}{2}} B + T^{-\frac{1}{2}} \mathbb{E}[\|\boldsymbol{\epsilon}\|] \big) + \sigma T^{-\frac{1}{2}} \Big)$$

$$= O\Big( T^{-\frac{\nu}{d}-1} + \sigma T^{-\frac{1}{2}} \Big),$$

which yields Corollary 1.

For the misspecified setting, we substitute (57)–(58) into (52), and obtain

$$\mathbb{E}\Big[|I - \hat{I}_1 - \hat{R}|\Big] = O\Big( \sqrt{p_{\max}} T^{-\frac{1}{2}} \big( T^{-\frac{\min(\nu^-,\nu)+\nu^+-\nu-\nu^++\nu}{d}-\frac{1}{2}} B + T^{-\frac{\nu+\nu^--\nu^+}{d}-\frac{1}{2}} B + T^{-\frac{1}{2}} \mathbb{E}[\|\boldsymbol{\epsilon}\|] \big) + \sigma T^{-\frac{1}{2}} \Big)$$

$$= O\Big( T^{-\frac{\min(\nu^-,\nu)}{d}-1} + T^{-\frac{\nu+\nu^--\nu^+}{d}-1} + \sigma T^{-\frac{1}{2}} \Big),$$

which yields Corollary 2.

### E.3 Proof of Corollary 3 (Noiseless SE Upper Bound)

Related to the fill distance discussed in the previous subsection, it has been shown in (Wendland, 2004) that the convergence orders for infinitely smooth functions turn out to depend exponentially on the fill distance $h_X$. In particular for SE kernel, the $L^2$ guarantee is illustrated in the following lemma.

**Lemma 9.** (Noiseless $L^2$ Guarantee (Wendland, 2004, Thm. 11.12)) *Let $f$ be any function in $\mathcal{H}_{\mathrm{SE}}$, there exist constants $h_0, C_s, \widetilde{C_s}$ such that $\forall X \subseteq D$ with $h_X \leq h_0$, and when $\mu_T$ is calculated with $X$ and $\lambda \leq \exp\left(\frac{2(C_s \log(h_X) - \widetilde{C_s})}{h_X}\right)$, the $L^2$ approximation error is bounded by*

$$\|f - \mu_T\|_{L^2} = O\left(e^{C_s \log(h_X)/h_X}\right) \tag{59}$$

As opposed to the Matérn kernel , the scaling of $h_X$ induced via maximum variance sampling for SE kernel functions is not clear; see the discussion in (Santin & Haasdonk, 2017, Sec. 4.1). To overcome this, we adopt the simpler approach of sampling on a uniformly-spaced grid, i.e., $X = \{(\frac{k_1}{N}, \ldots, \frac{k_d}{N})|k_i \in \{0, \ldots, N-1\}\}$ with $N = T^{\frac{1}{d}}$, to ensure that $h_X = T^{-\frac{1}{d}}$ (e.g., see (Kanagawa & Hennig, 2019, Thm. 4.3)). The substitution of $h_X = T^{-\frac{1}{d}}$ into (59) then yields

$$\|f - \mu_T\|_{L^2} = O\left(T^{-\frac{C_s}{d}T^{-\frac{1}{d}}}\right).$$

By applying Theorem 2 with $\sigma = 0$ and $\lambda = 0$ (Lemma 9 still holds), the proof of Corollary 3 is complete.

### E.4 Proof of Corollary 4 (Noisy SE Upper Bound)

The bulk of this subsection is devoted to introducing definitions and results from (Bach, 2017). Although we are working towards the noisy setting, all results stated are for the noiseless setting until stated otherwise.

It is often useful to study an RKHS through an integral operator $\Sigma$, which leads to an isometry with $L^2(d\rho)$ space with measure $d\rho$:

$$(\Sigma f)(\cdot) = \int_{\mathcal{X}} f(\mathbf{x})k(\mathbf{x}, \cdot)d\rho(\mathbf{x}).$$

For the moment, we consider any kernel (including SE) that can be written in the form

$$k(\mathbf{x}, \mathbf{x}') = \int \phi(\mathbf{x}, \boldsymbol{\omega})\phi(\mathbf{x}', \boldsymbol{\omega})d\rho(\boldsymbol{\omega}),$$

for some $\phi(\mathbf{x}, \cdot) : L^2(d\rho) \to L^2(d\rho)$. Let $\{\mathbf{x}_i\}_{i=1}^T$ be i.i.d samples drawn from density $q$ with respect to measure $d\rho$, and pick weights $\boldsymbol{\beta}$ such that the approximation of $f$ is

$$\hat{f}(\cdot) = \sum_{i=1}^T \frac{\beta_i}{\sqrt{q(\mathbf{x}_i)}}\phi(\mathbf{x}_i, \cdot),$$

which belongs to the Hilbert space $\hat{\mathcal{H}}^k$ formed by the approximated kernel $\hat{k}$ through $T$ random features $\{\boldsymbol{\omega}_i\}_{i=1}^T$ with the same density $q$:

$$\hat{k}(\mathbf{x}, \mathbf{y}) = \frac{1}{T}\sum_{i=1}^T \frac{1}{q(\boldsymbol{\omega}_i)}\phi(\mathbf{x}, \boldsymbol{\omega}_i)\phi(\mathbf{y}, \boldsymbol{\omega}_i).$$

The weights $\boldsymbol{\beta}$ are chosen to solve the following minimization problem with Lagrange multiplier $\lambda > 0$:

$$\min_{\beta} \|f - \hat{f}\|_{L^2(d\rho)} + T\lambda\|\boldsymbol{\beta}\|^2,$$

with the solution

$$\boldsymbol{\beta} = \frac{1}{T}\Phi^T\left(\frac{1}{T}\Phi\Phi^T + \lambda\mathbf{I}\right)^{-1}f, \quad \|\boldsymbol{\beta}\|^2 \leq \frac{4}{T} \tag{60}$$

where $\Phi : \mathbb{R}^T \to L^2(d\rho)$ is an operator:

$$\Phi\boldsymbol{\beta} = \sum_{i=1}^T \frac{\beta_i}{\sqrt{q(\boldsymbol{\omega}_i)}}\phi(\boldsymbol{\omega}_i, \cdot).$$

The quantity $\Phi\Phi^T$ in turn defines th following empirical integral operator $\hat{\Sigma}: L^2(d\rho) \to L^2(d\rho)$:

$$(\hat{\Sigma}f)(\cdot) = \frac{1}{T}\sum_{i=1}^{T}\frac{1}{q(\boldsymbol{\omega}_i)}\langle f, \phi(\boldsymbol{\omega}_i, \cdot)\rangle_{L^2(d\rho)}\phi(\boldsymbol{\omega}_i, \cdot).$$

This allows us to write $\hat{f}$ as

$$\hat{f} = \Sigma^{1/2}\hat{\Sigma}(\hat{\Sigma} + \lambda I)^{-1}\Sigma^{-1/2}f, \tag{61}$$

where $\Sigma^{1/2}$ is the unique positive self-adjoint square root of $\Sigma$, and forms a bijection from $L^2(d\rho)$ to the RKHS $\mathcal{H}_k$ (Bach, 2017).

With the above definitions, it was shown in (Bach, 2017) that the noiseless $L^2$-error between $f$ and $\hat{f}$ corresponds to the eigenvalue decay of the integral operator $\Sigma$ if $q$ is the optimized distribution:

$$q(\mathbf{x}) \propto \sum_{i\geq 1}\frac{\mu_i}{\mu_i + \lambda}e_i(\mathbf{x})^2, \tag{62}$$

where $\mu_i$ is the $i$-th largest eigenvalue of $\Sigma$, and $e_i(\mathbf{x})$ is the corresponding eigenfunction. Specifically, $\|f - \hat{f}\|_{L^2(d\rho)}$ has a geometric error (e.g., $\exp(-i^{\frac{1}{d}})$) if $\mu_i$ decays geometrically/exponentially. The guarantee that we make use of is formally stated as follows.

**Lemma 10.** (Bach, 2017, Prop. 2) *For the optimized distribution defined in* (62)*, and the estimate $\hat{f}$ defined in* (61)*, let $\delta > 0$ and $d_\lambda = \text{Tr}\Sigma(\Sigma + \lambda\mathbf{I})^{-1}$, and assume that $T \geq 5d_\lambda\log(\frac{16d_\lambda}{\delta})$. Then, it holds with probability at least $1 - \delta$ that*

$$\inf_{\|f\|_{\mathcal{H}_k}\leq 1}\sup_{\|\hat{f}\|_{\hat{\mathcal{H}}^k}\leq 2}\|f - \hat{f}\|_{L^2(d\rho)} \leq 2\sqrt{\lambda}.$$

Note that $d_\lambda$ represents a notion of *effective dimension* or *effective degrees of freedom*. For the SE kernel integral operator, the eigenvalue decay satisfies the following (e.g., see (Santin & Schaback, 2016, Thm. 15)):

$$\mu_i = O\big(\exp(-C_e i^{\frac{1}{d}})\big), \tag{63}$$

for some constant $C_e > 0$. Moreover, (Sun et al., 2018, Lem. 6) shows that, for $\mu_i$ decaying according to (63), it holds that $d_\lambda = O((\log\frac{1}{\lambda})^d)$. Rearranging $T = \Theta(d_\lambda\log(d_\lambda))$ in Lemma 10 gives us that

$$d_\lambda = \Theta\Big(\frac{T}{\log T}\Big).$$

Then, by equating $\Theta\big(\frac{T}{\log T}\big)$ with $(\log\frac{1}{\lambda})^d$ and rearranging, we obtain

$$\sqrt{\lambda} = O\bigg(\exp\Big(-C_r\Big(\frac{T}{\log T}\Big)^{\frac{1}{d}}\Big)\bigg) \tag{64}$$

for some constant $C_r > 0$. This determines the $L^2$-error between $f$ and $\hat{f}$ in Lemma 10 in the absence of noise.

To account for the effect of noise, we use the following lemma to extend Lemma 10.

**Lemma 11.** (Bach, 2017, Sec. 5) *Under the preceding setup, if each function query is corrupted by independent noise with variance not exceeding $q(\mathbf{x}_i)\sigma^2$ in the $i$-th entry, then we have*

$$\inf_{\|f\|_{\mathcal{H}_k}\leq 1}\sup_{\|\hat{f}\|_{\hat{\mathcal{H}}^k}\leq 2}\mathbb{E}\big[\|f - \hat{f}\|_{L^2(d\rho)}\big] \leq 2\sqrt{\lambda} + \sigma\|\boldsymbol{\beta}\|.$$

While the presence of $q(\mathbf{x}_i)$ in the preceding statement seems complicated, it is fortunately considerably simplified in our case due to the following.

**Lemma 12.** (Bach, 2017, Sec. 4.4) *For shift-invariant kernels in $[0,1]^d$ (including SE), the optimized distribution $q$ is the uniform distribution when the uniform measure $d\rho$ is used.*

Therefore, we have $q(\cdot) = 1$ (i.e., the uniform distribution over $[0,1]^d$), and the additional noisy term in Lemma 11 is at most $\sigma\|\boldsymbol{\beta}\| \leq 2\sigma T^{-\frac{1}{2}}$ due to the upper bound on $\|\boldsymbol{\beta}\|$ given in (60). Combining with (64), the noisy $L^2$ guarantee becomes

$$\inf_{\|f\|_{\mathcal{H}_k}\leq 1} \sup_{\|\hat{f}\|_{\mathcal{H}^k}\leq 2} \mathbb{E}\big[\|f - \hat{f}\|_{L^2(d\rho)}\big] = O\bigg(\exp\Big(-C_r\Big(\frac{T}{\log T}\Big)^{\frac{1}{d}}\Big) + \sigma T^{-\frac{1}{2}}\bigg),$$

and substituting into our general BQ guarantee (Theorem 2), we obtain Corollary 4.

# F   Alternative Analysis Based on Confidence Bounds

In this section, we present an analysis based on constructing confidence intervals of the true function on each time step, which is a popular strategy in the analysis Bayesian optimization (BO) algorithms. Our analysis is most closely related to that of the BO simple regret in (Vakili et al., 2021a).

## F.1   Noisy Setting

In this subsection, we present our analysis in a slightly different way than our earlier results. Specifically, we first state the results with high probability (i.e., $1 - \delta$), and then convert them to expectations in the same manner as done in (Vakili et al., 2021a, App. F), by setting $\delta = \frac{1}{\sqrt{T}}$.

We first provide several existing results that we use to prove results in (12) and (14).

**Lemma 13.** (Lee et al., 2022, Prop. 1 & Remark 5) *Let $\mathcal{H}_k$ be the set of functions whose RKHS norm is upper bounded by a constant $B > 0$. Then $f$ is $L$-Lipschitz continuous with some constant $L$ depending only on the kernel parameters.*

In the following, we use the shorthand notation $\mathbf{x}_{1:T} = (\mathbf{x}_1, \ldots, \mathbf{x}_T)$, and similarly for other quantities indexed by $t$. Recall also the posterior mean and variance defined in (4)–(5), with parameter $\lambda > 0$.

**Lemma 14.** (Confidence Intervals (Vakili et al., 2021a, Thm. 1)) *Fix a function $f$ satisfying $\|f\|_{\mathcal{H}_k} \leq B$, and assume Gaussian noises with variance $\sigma^2$. Assume further that $\mathbf{x}_{1:T}$ are independent of $\epsilon_{1:T}$, i.e., the points are chosen non-adaptively. For a fixed $\mathbf{x} \in D$, for any $t \in [T]$, define the upper and lower confidence bounds as*

$$U_t^\delta(\mathbf{x}) = \mu_t(\mathbf{x}) + (B + \beta(\delta))\sigma_t(\mathbf{x}),$$
$$L_t^\delta(\mathbf{x}) = \mu_t(\mathbf{x}) - (B + \beta(\delta))\sigma_t(\mathbf{x}),$$

*with $\beta(\delta) = \frac{\sigma}{\lambda}\sqrt{2\log\frac{1}{\delta}}$, and $\delta \in (0,1)$. Then, we have for any $\mathbf{x} \in D$ that*

$$f(\mathbf{x}) \leq U_t^\delta(\mathbf{x}) \quad \text{w.p. at least } 1 - \delta \tag{65}$$
$$f(\mathbf{x}) \geq L_t^\delta(\mathbf{x}) \quad \text{w.p. at least } 1 - \delta. \tag{66}$$

**Lemma 15.** (Adaption of (Srinivas et al., 2010, Lem. 5.4)) *Letting $\gamma_T = \sup_{x_{1:T}\subseteq D} I(y_{1:T}; f_{1:T})$, where $f_{1:T} = (f(\mathbf{x}_{1:T})) \in \mathbb{R}^T$ denotes the function values at the points $\mathbf{x}_1, \ldots, \mathbf{x}_T$, we have*

$$\sum_{t=1}^T \sigma_{t-1}^2(\mathbf{x}_t) \leq \frac{2\gamma_T}{\log\big(1 + \frac{1}{\lambda^2}\big)}. \tag{67}$$

Let $\widetilde{D}$ be a finite subdomain of $D = [0,1]^d$ with $T^{d/2}$ points, with equal spacing of width $\frac{1}{\sqrt{T}}$ in each dimension. For any $\mathbf{x} \in D$, let $[\mathbf{x}]_{\widetilde{D}} = \arg\min_{\mathbf{x}'\in\widetilde{D}} \|\mathbf{x} - \mathbf{x}'\|_2$. By construction, we have, for any $\mathbf{x} \in D$ that

$$\big\|\mathbf{x} - [\mathbf{x}]_{\widetilde{D}}\big\|_2 \leq \frac{\sqrt{d}}{\sqrt{T}} = O\Big(\frac{1}{\sqrt{T}}\Big). \tag{68}$$

By Lemma 13, the function $f$ is $L$-Lipschitz. Thus we have for any $\mathbf{x} \in D$ that

$$|f(\mathbf{x}) - f([\mathbf{x}]_{\widetilde{D}})| \leq L\|\mathbf{x} - [\mathbf{x}]_{\widetilde{D}}\|_2 = O\left(\frac{L}{\sqrt{T}}\right). \tag{69}$$

For any fixed $\mathbf{x} \in \widetilde{D}$, applying Lemma 14 gives the following with probability at least $1 - \frac{\delta}{2|\widetilde{D}|}$:

$$f(\mathbf{x}) \geq \mu_T(\mathbf{x}) - \left(B + \beta\left(\frac{\delta}{|\widetilde{D}|}\right)\right)\sigma_T(\mathbf{x}). \tag{70}$$

By a union bound over all $\mathbf{x} \in \widetilde{D}$, we have, for all $\mathbf{x} \in \widetilde{D}$ simultaneously that

$$f(\mathbf{x}) \geq \mu_T(\mathbf{x}) - \left(B + \beta\left(\frac{\delta}{|\widetilde{D}|}\right)\right)\sigma_T(\mathbf{x}), \tag{71}$$

with probability at least $1 - \frac{\delta}{2}$. Similarly, we have, for all $\mathbf{x} \in \widetilde{D}$ that

$$f(\mathbf{x}) \leq \mu_T(\mathbf{x}) + \left(B + \beta\left(\frac{\delta}{|\widetilde{D}|}\right)\right)\sigma_T(\mathbf{x}), \tag{72}$$

with probability at least $1 - \frac{\delta}{2}$. Combining (71) and (72), and again applying the union bound, we have, for all $\mathbf{x} \in \widetilde{D}$ that

$$|f(\mathbf{x}) - \mu_T(\mathbf{x})| \leq \left(B + \beta\left(\frac{\delta}{|\widetilde{D}|}\right)\right)\sigma_T(\mathbf{x}), \tag{73}$$

with probability at least $1 - \delta$. We now extend the above upper bound to any point $\mathbf{x}$ in the domain $D$:

$$\begin{aligned}
|f(\mathbf{x}) - \mu_T(\mathbf{x})| &\leq \left|f(\mathbf{x}) - f([\mathbf{x}]_{\widetilde{D}}) + f([\mathbf{x}]_{\widetilde{D}}) - \mu_T([\mathbf{x}]_{\widetilde{D}}) + \mu_T([\mathbf{x}]_{\widetilde{D}}) - \mu_T(\mathbf{x})\right| \\
&\leq \left|f(\mathbf{x}) - f([\mathbf{x}]_{\widetilde{D}})\right| + \left|f([\mathbf{x}]_{\widetilde{D}}) - \mu_T([\mathbf{x}]_{\widetilde{D}})\right| + \left|\mu_T([\mathbf{x}]_{\widetilde{D}}) - \mu_T(\mathbf{x})\right| \\
&\leq O\left(\frac{L}{\sqrt{T}}\right) + \left(B + \beta\left(\frac{\delta}{|\widetilde{D}|}\right)\right)\sigma_T([\mathbf{x}]_{\widetilde{D}}) + O\left(\frac{L}{\sqrt{T}}\right),
\end{aligned} \tag{74}$$

where the second inequality is by applying the triangle inequality, and the third inequality is due to (73) and (69).

We will now show an upper bound on the posterior variance $\sigma_T(\mathbf{x})$. Due to the decreasing property of posterior variance, we know that $\sigma_{t+1}(\mathbf{x}) \leq \sigma_t(\mathbf{x})$ for all $\mathbf{x}$ and $t$. Furthermore, due to the maximum variance sampling strategy, we have $\sigma_{t-1}(\mathbf{x}_t) \geq \sigma_{t-1}(\mathbf{x})$. Thus, we have

$$\sigma_T([\mathbf{x}]_{\widetilde{D}}) \leq \sigma_{t-1}([\mathbf{x}]_{\widetilde{D}}) \leq \sigma_{t-1}(\mathbf{x}_t) \tag{75}$$

for all $\mathbf{x}$ and $t \leq T$. Squaring and averaging over $t \in [T]$ gives

$$\sigma_T^2([\mathbf{x}]_{\widetilde{D}}) \leq \frac{1}{T}\sum_{t=1}^{T}\sigma_{t-1}^2(\mathbf{x}_t), \tag{76}$$

and applying Lemma 15, we obtain

$$\sigma_T([\mathbf{x}]_{\widetilde{D}}) \leq \sqrt{\frac{2\gamma_T}{T\log\left(1 + \frac{1}{\lambda^2}\right)}}. \tag{77}$$

Substituting (77) and $|\widetilde{D}| = T^{d/2}$ into (74), and recalling the definition of $\beta(\cdot)$ in Lemma 14, we obtain

$$|f(\mathbf{x}) - \mu_T(\mathbf{x})| \leq O\left(\frac{L}{\sqrt{T}}\right) + \left(B + \frac{\sigma}{\lambda}\sqrt{d\log T + 2\log\frac{1}{\delta}}\right)\sqrt{\frac{2\gamma_T}{T\log\left(1 + \frac{1}{\lambda^2}\right)}} \tag{78}$$

$$= O\left(\sqrt{\frac{\gamma_T}{T}}\left(B + \sigma\sqrt{\left(d\log T + \log\frac{1}{\delta}\right)}\right)\right). \tag{79}$$

Finally, the absolute error of the above algorithm can be upper bounded by

$$\left| \int_D p(\mathbf{x})(f(\mathbf{x}) - \mu_T(\mathbf{x}))d\mathbf{x} \right| \leq \max_{\mathbf{x} \in D} \left| f(\mathbf{x}) - \mu_T(\mathbf{x}) \right| \int_D p(\mathbf{x})d\mathbf{x} = O\left( \sqrt{\frac{\gamma_T}{T}} \left( B + \sigma \sqrt{\left( d \log T + \log \frac{1}{\delta} \right)} \right) \right). \tag{80}$$

with probability at least $1 - \delta$. For the Matérn-$\nu$ kernel, $\gamma_T = O\left( T^{\frac{d}{2\nu+d}} (\log T)^{\frac{2\nu}{2\nu+d}} \right)$, and for the SE kernel, $\gamma_T = O\left( (\log(T))^{d+1} \right)$ (see (Vakili et al., 2021b)). Thus, we obtain the noisy upper bounds in (12) and (14) upon setting $\delta = \frac{1}{\sqrt{T}}$ as outlined at the start of this subsection.

### F.2 Noiseless Setting

For the noiseless setting, we first state two useful lemmas, the first giving a standard deterministic confidence bound, and the second relating the posterior variance and the fill distance $h_X$ (see (53)).

**Lemma 16.** (Kanagawa et al., 2018, Cor. 3.11) *For any $f \in \mathcal{H}_k$ with $\|f\|_{\mathcal{H}_k} \leq B$, it holds that $L_t(\mathbf{x}) \leq f(\mathbf{x}) \leq U_t(\mathbf{x})$ for any $t$ and $\mathbf{x} \in D$, where*

$$U_t(\mathbf{x}) = \mu_{t-1}(\mathbf{x}) + B\sigma_{t-1}(\mathbf{x}),$$
$$L_t(\mathbf{x}) = \mu_{t-1}(\mathbf{x}) - B\sigma_{t-1}(\mathbf{x}),$$

*and where $\mu_{t-1}(\cdot)$ and $\sigma_{t-1}(\cdot)$ are given in (4)–(5) and calculated with $\lambda = 0$.*

**Lemma 17.** (Adaption of (Santin & Haasdonk, 2017, Thm. 3)) *Consider the set of points $X := \{\mathbf{x}_t\}_{t=1}^T$ obtained by Algorithm 2. Then:*

1. *For functions in the Matérn RKHS, we have*

$$\max_{\mathbf{x} \in D} \sigma_T(\mathbf{x}) = O(h_X^\nu). \tag{81}$$

2. *For functions in the SE RKHS, there exists $C_p > 0$ such that*

$$\max_{\mathbf{x} \in D} \sigma_T(\mathbf{x}) = O\left( e^{-\frac{C_p}{h_X}} \right). \tag{82}$$

For the Matérn kernel, as discussed in Appendix E.2, it is known that maximum-variance sampling leads to the fill distance being $h_X = \Theta(T^{-\frac{1}{d}})$, which is the best possible. Hence, the desired upper bound (13) for the Matérn kernel follows directly from (81) along with Lemma 16 and similar steps to (78)–(80).

As mentioned in Appendix E.3, the fill distance of MVS on the SE kernel is not clear. To facilitate a similar treatment, one may adopt the same grid points in (82) to derive an upper bound of $O(e^{-C_p T^{-\frac{1}{d}}})$. However, a more recent result offers a more explicit dependence on the constant using grid points, as demonstrated below.

**Lemma 18.** ((Xu et al., 2022, Eq. (32))) *Consider the domain $D = [0,1]^d$ and a grid-based subset $X \subset D$. Then, sampling each point once yields the following upper bound on the noiseless posterior standard deviation (with $\lambda = 0$):*

$$\max_{\mathbf{x} \in D} \sigma_T(\mathbf{x}) = O\left( e^{-\frac{d}{2} T^{\frac{1}{d}}} \right). \tag{83}$$

Then, combining with Lemma 16 and proceeding similarly to (80), we obtain the noiseless upper bound in (15).

## G Additional Experimental Results

### G.1 Further Synthetic Experiments

In Figures 4, 5, and 6, we present additional results on synthetic kernel-based functions and benchmark functions. Overall, while there is no definitive ordering between the methods in general, we observe similar

findings to those discussed in Section 5.2. In particular, for the plots of error vs. split fraction, the trend can be decreasing (particularly at low noise), increasing (particularly at high noise), or "U-shaped", but 0.5 is generally a reasonable choice. As we already discussed in Section 5.2, there is often a sudden drop at 1.0.

We note that some of the MVS curves exhibit non-monotone behavior (e.g., for Alpine-2D). We believe that this is because the only randomness in MVS is in the noise and the 3 initial points, whereas MC and MVS-MC have much more randomness due to being randomized algorithms. When there is limited randomness and few queries have been made, the algorithm is essentially outputting an uncertain guess, and it can happen that this guess luckily has a low error, but then this luck diminishes as more samples are taken. In contrast, MVS-MC and MC have enough internal randomness to "average out" the lucky and unlucky scenarios.

### G.2 Sensor Measurement Data

We consider the problem of estimating an average sensor reading from limited queries, which each query consists of reading the value at a given time instant. Note that since the algorithms we consider are non-adaptive, the query times can be pre-computed. The data set consists of energy consumption readings for London Households that took part in the UK Power Networks led Low Carbon London Project, between November 2011 and February 2014.[8]

We construct a time-series signal (shown in Figure 7) of length 19,548 by sampling the data at intervals of one hour. Our goal is to estimate the average energy consumption during this period. Although the domain is now discrete, we let MVS and MVS-MC work on the continuous space, and round the selected decimal value to the nearest point in the data set. To create a *noisy* BQ problem, we artificially add Gaussian $N(0, \sigma^2)$ noise to each query, with $\sigma \in \{0, 0.1, 0.5\}$. The results are shown in Figure 8; in this case, we found the various methods to perform relatively similarly to each other.

---

[8]The data can be downloaded at data.london.gov.uk.

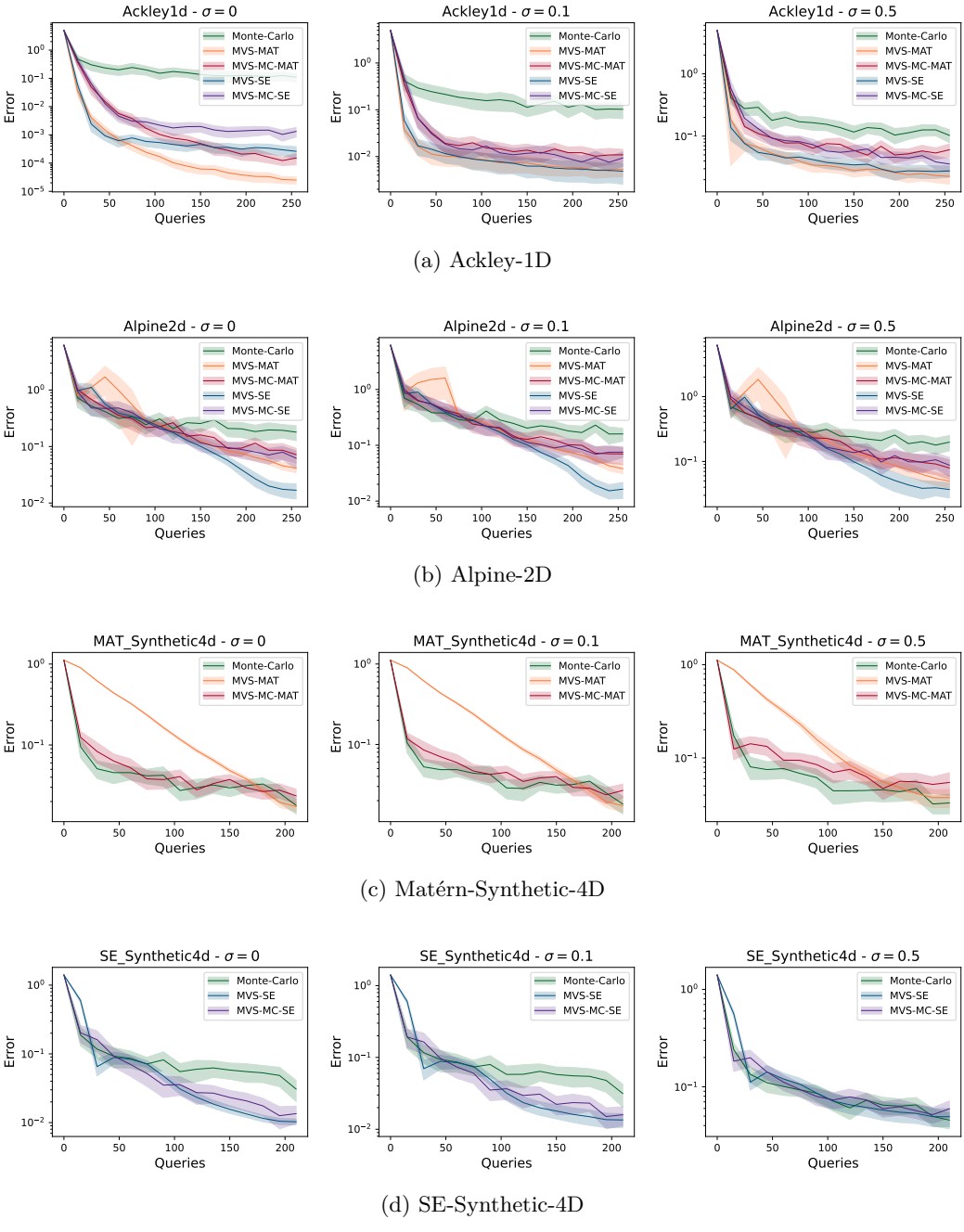

Figure 4: Comparison of algorithms and the effect of noise.

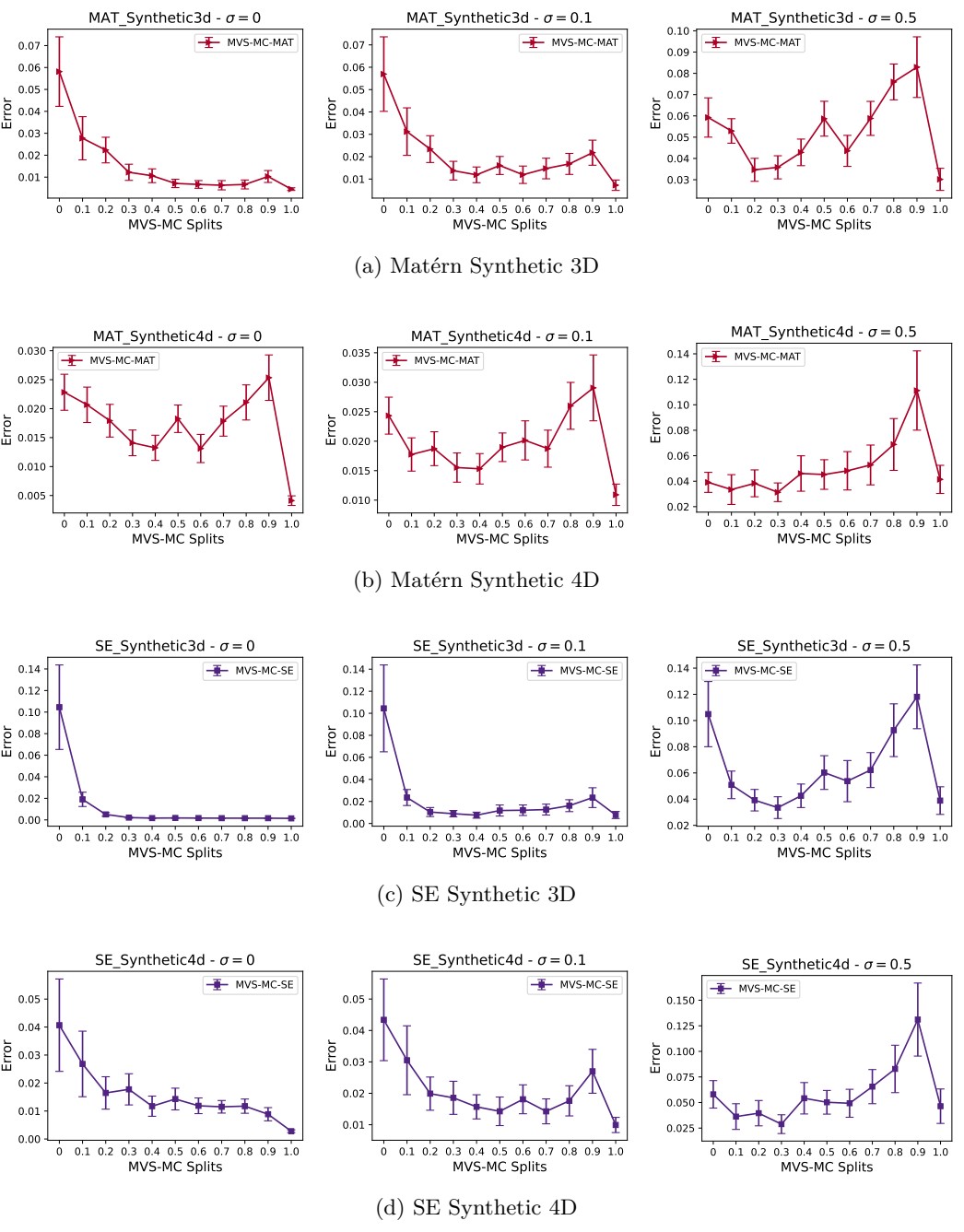

Figure 5: Comparison of different MVS-MC splits, i.e., the fraction of rounds for which MVS is used.

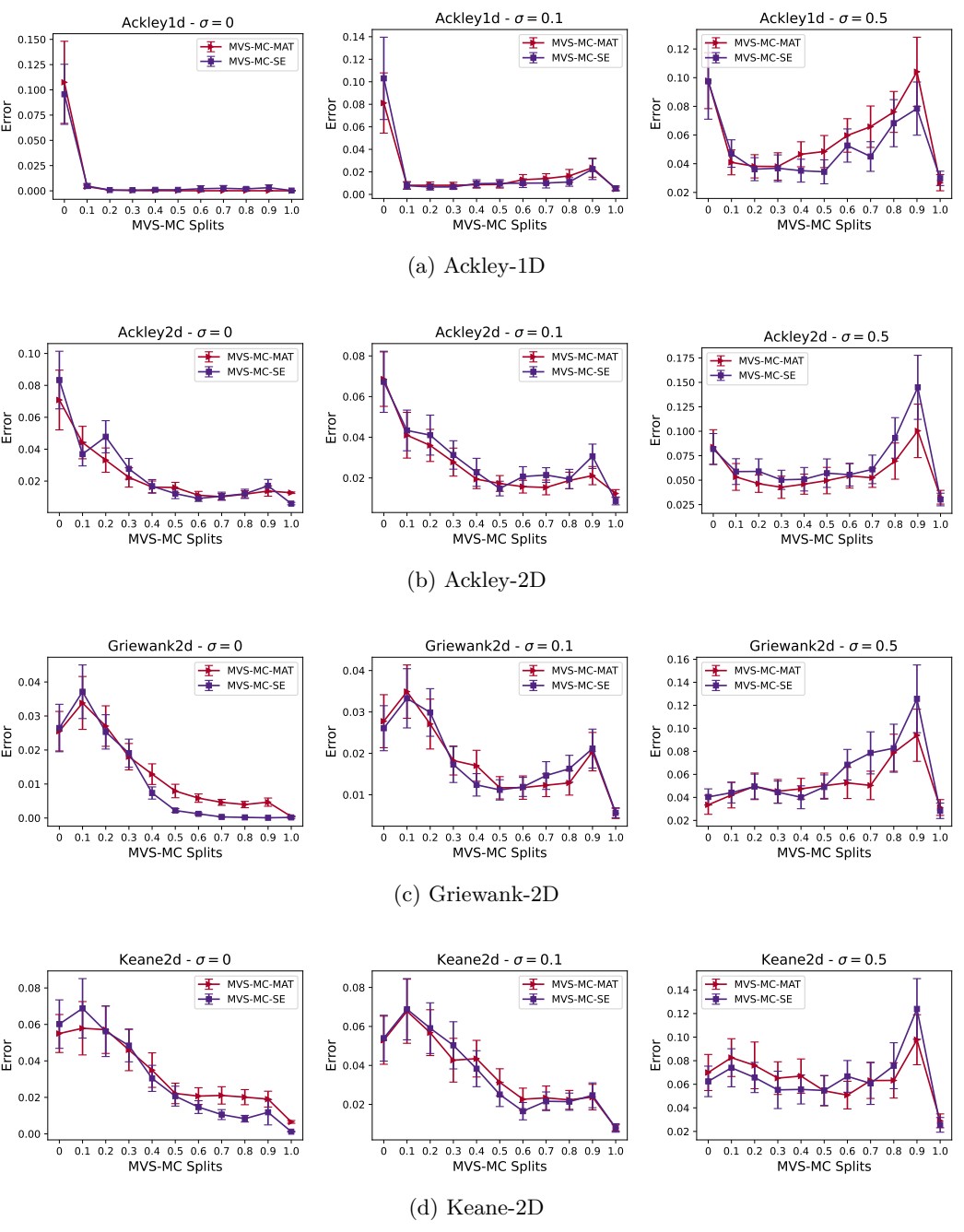

Figure 6: Comparison of different MVS-MC splits, i.e., the fraction of rounds for which MVS is used.

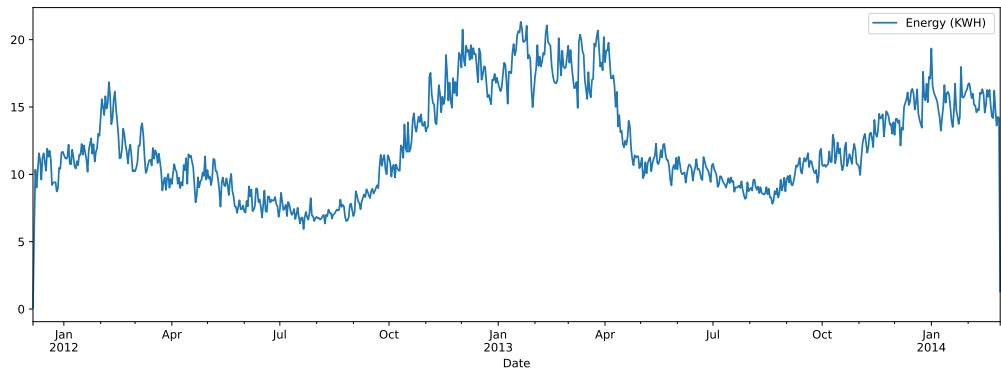

Figure 7: Time-series function for energy measurements.

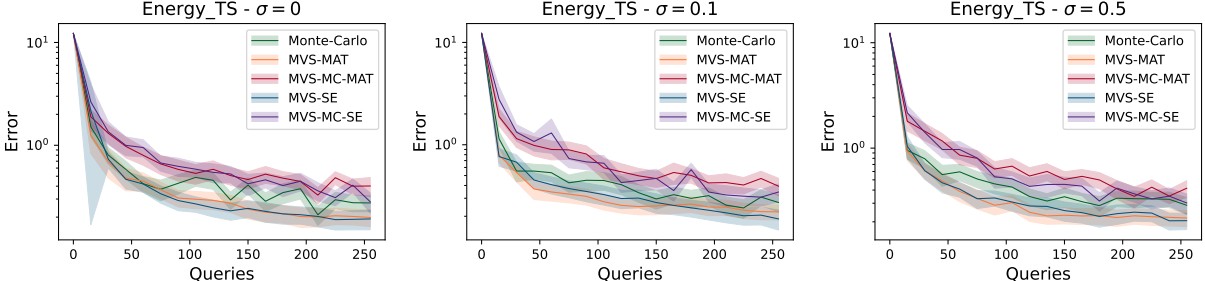

Figure 8: Results for time series energy data.

