# OpenReview forum: "On Average-Case Error Bounds for Kernel-Based Bayesian Quadrature"
_TMLR — Accepted by TMLR_

### Review · Reviewer_towK · 2023-04-01

**Summary Of Contributions:**

This paper proves upper and lower bounds on the average-case error of randomized numerical integration in RKHSs of Matérn kernels (i.e., Sobolev spaces) and in the RKHS of the squared exponential (SE) kernel, both for noisy and noiseless integrand evaluations. Upper bounds are proved by the means of a two-step "meta-algorithm" that uses maximum variance sampling to select half the samples and random sampling to select the remaining half.

The bounds for Matérns are not new, having been proven by L. Plaskota in 1996, but the authors claim to supply new proofs that may have some advantages over those by Plaskota.

For the SE kernel only upper bounds are proved. These appear to be new, though, as I will detail later, there are better bounds in the literature.


**Audience:**

Yes

**Broader Impact Concerns:**

-

**Claims And Evidence:**

Yes

**Requested Changes:**

Critical:

1. Better bounds for the SE kernel are available in the literature. From Eq (11.11) in Theorem 11.22 of H. Wendland's 2005 book "Scattered Data Approximation" (a fairly standard source of BQ upper bounds) one easily obtains a noiseless worst-case upper bound of the form $\exp(-C T^{1/d} \log T)$, which is better than any bound in this paper. Wendland's Theorem 11.22 can also be used to improve Lemma 17 in Appendix F and the results that exploit this lemma. The only occurrence of the aforementioned bound in the BQ literature that I am aware of is as Theorem 2.20 in the 2019 thesis "Kernel-Based and Bayesian Methods for Numerical Integration" by T. Karvonen. A more explicit upper bound is probably obtainable by combining Eq (1.5) in Theorem 1.2 of https://arxiv.org/abs/2209.12473 with a basic tensor-product construction.

2. In appendices, there are substantial issues in how the Fourier definition of the Sobolev norm is used. In Eqs. (18) and (19), the norm is defined by integrating over $D$ (an issue that is also present in the integrals on p16). This is not the correct way to proceed. Instead, one has to consider restrictions on $D$ of functions defined on $\mathbb{R}^d$ (to define the Fourier transform you need a function on the whole real space rather than only on its subset). See e.g. Section 3.1.1 in Teckentrup (2020).

3. Perhaps I have missed something here, but I think that Wenzel et al. (2021) only consider the case $\lambda = 0$, which would mean that their results cannot be used (at least without modification) to prove Corollaries 1 and 2 in Appendix E.2. [Moreover, p24 refers to Theorem 11 in this paper but there is no such theorem in the paper - there is Lemma 11 though but that is not about the fill-distance.]

4. From Appendix B one gets the impression that the authors are claiming a generalisation to fractional Sobolev spaces of the standard lower bounds for integration in Sobolev spaces. Such fractional versions are well known (at least for the worst-case error, and I would be very surprised if things were different for the average-case error). I do not recall what a standard reference for this is, but see at least Proposition 8 on p143 of K. Ritter's 2000 book "Average-Case Analysis of Numerical Problems". For approximation (as opposed to integration) the most general treatment is given in E. Novak & H. Triebel "Function Spaces in Lipschitz Domains and Optimal Rates of Convergence for Sampling", Constructive Approximation, 2006.

5. The paper is full of references to Novak (1988). Most of these references are for specific facts about convergence rates etc. and should therefore provide more specific pointers than the entire book (e.g., theorem number).


Other:
- p2: "maximxum"
- p3: "...where the ABQ algorithm only achieves $O(T^{-\nu/d})$ scaling" I believe this is only because the proof technique that Kanagawa and Hennig use is sub-optimal.
- p3: "are have"
- p6: What is $\mu_0$ in the input for Algorithm 2?
- p7: ". we specifically..."
- p8: "...note that that the techniques..."
- p13: In Appendix A it is stated that the norms in Eqs (17) and (19) are equal. In reality these define norms that are equivalent, which one sees by applying the binomial theorem to the polynomial in the Fourier transform and using the properties of the Fourier transform; after this it is seen that each $L^2$-norm of a weak derivative in Eq (19) is multiplied by a binomial coefficient not present in Eq (17).
- p16: What are $\mathcal{H}^{n - \frac{d}{2}}$ etc. in Eqs. (25)-(30)?
- p28: "usefull"
- p28: I think there should not be "with probability one" in the statement of Lemma 15.


**Strengths And Weaknesses:**

As detailed in the paper, little theoretical work on Bayesian quadrature (BQ) in a noisy setting exists, so any contribution to this direction would obviously be welcome. It is therefore a shame that the connection of the paper to Bayesian quadrature is tenuous at best. The paper is really about fundamental limits of computation in certain function spaces when the integrand evaluations are noisy than rather than error bounds for some commonly used Bayesian quadrature method in the noisy setting (the lower bounds obviously apply to such "standard" BQ methods as well). That is, the upper bounds apply to a certain two-step "meta-algorithm" that combines standard BQ with a Monte Carlo residual, but I have never seen this method used in the BQ literature. It seems that the numerical experiments in Section 5 do not show this algorithm outperforms standard BQ (I take it that MVS-SE and MVS-MAT refer to standard BQ, i.e. integration of Eq (4) with the integration points obtained from maximum variance sampling).

It is good that the authors are explicit that the lower bounds for Matérns have been derived by Plaskota a while ago and that they merely provide new proofs. Although there are statements here and there about how the proofs differ from those by Plaskota, one is left wondering how significant these new proofs really are: Can the technique yield new results in some cases? Is it simpler or more general?

The paper is generally reasonably well written. However, there are many issues with mathematics, both real and presentational.

---

> ### Author Response · Authors · 2023-04-27
> **Response to Reviewer toWK**
>
> Thank you for your feedback – we found this review to be very helpful.  We respond to the main points as follows.
>
> Q. **Implications of the lower bound proof technique.**
>
> A.  While the lower bound for Matérn functions has been previously derived by Plaskota, we believe that there is value in identifying an explicit “hard instance” that captures both sources of difficulty simultaneously, rather than two separate simplifications to noiseless and constant functions.  While the latter approach also works in this particular setting, having multiple approaches may lead to a better ability to generalize to other settings.
>
> For example, consider integrating **$\exp(-\lambda f(x))$** instead of **$f(x)$**, and requiring a multiplicative **$1 \pm \epsilon$** guarantee.  As **$\lambda$** increases, the problem becomes more like optimization, for which we know the noiseless lower bound is not tight for noisy settings.  In such scenarios, we believe it would be necessary to build on a proof technique along the lines of our one.
>
> Q. **Better worst-case SE upper bound**
>
> A.  We were indeed aware of the existence of this bound, which has an additional log T term in the exponent (we have listed one on the worst-case lower bound in Table.1).  One reason that we chose to use the bound in Lemma 17 is since we prefer the explicit dependence on the constants (i.e., purely d/2 rather than C), and logarithmic factors are not an emphasis of the paper.
>     In addition, our intention is to demonstrate the generality of the two-batch algorithm in attaining a **$T^{-1/2}$** improvement over general worst-case upper bounds.  Hence, the mentioned better worst-case upper bound can also be applied to attain a corresponding average-case guarantee.
>     We have further highlighted Wendland’s book given that it is a common source of these bounds, e.g., see Corollary 3 on p8 and Appendix E.3 on p25.
>
>
> Q. **MVS is only known to lead to grid-like points with $\lambda=0$.**
>
> A.  We thank the reviewer for pointing this out.  We have modified the algorithm to use **$\lambda=0$** for sampling, while still using the original **$\lambda$** for the final posterior mean formed.  We also added Footone 3 noting that other choices like a uniform grid are also sufficient.
>
> Q. **Fractional Sobolev lower bounds already exist.**
>
> A.  We appreciate the reviewer for providing explicit citations on the more general treatment of the extension to fractional Sobolev spaces.  We have added the citation and relevant discussion (below Theorem 3, p15) and avoided depicting it as new.
>
> Q. **In Appendix A it is stated that the norms in Eqs (17) and (19) are equal.**
> A.  We didn’t intend to depict these terms as equal, though we acknowledge that using the same notation on the left-hand side was an abuse of notation.  We have modified explicitly mention this abuse of notation in Appendix A.  We prefer not to create more notation by renaming one of them, but we can do so if it is deemed essential.
>
> Q: **Incorrect Fourier definitions.**
>
> A.  Thanks very much for noting this.  These definitions should indeed be for **$\mathbb{R}^d$** followed by a definition for **$D$** that takes the infimum over all functions on **$\mathbb{R}^d$** that match the original function when restricted to **$D$**.  We have edited Appendix A accordingly. Fortunately this does not impact our analysis/results.   The only place we use these definitions in our analysis is in (25)--(32), but there we are only considering functions consisting of “bumps” whose support is a subset of **$D$**.  This means that the extension to **$\mathbb{R}^d$** trivially sets the function to zero outside **$D$** anyway, so integrating over **$D$** or over **$\mathbb{R}^d$** makes no difference.
>
> Q. **There shouldn’t be "with probability one" in the statement of Lemma 15.**
>
> A.  If the data set defining the posterior is fixed/deterministic, then indeed there is no randomness so the phrase “with probability one” can be omitted.  For a random data set, we believe it should be kept.  We have reworded to clarify this, and have added a citation to Corollary 3.1.1 in “Gaussian processes and kernel methods: A review on connections and equivalences”.
>
> Q. **What is H^{n-d/2} in (25)-(30).**
>
> A.  It denotes the Matern RKHS with **$\nu$** being **$n-d/2$**, transforming from the Sobolev space **$W_2^n$** using norm equivalence in Lemma 1.  Note that **$n\ge 1$** is an integer, which ensures that the Matern RKHS is well-defined.  We have added a line to clarify this, and replaced the superscripts by subscripts for consistency.
>
> Q: **Other suggestions:**
>
> A: We have made the following modifications in view of the remaining comments:
> - Fixed the reference to Wenzel (2019)
> - Gave specific numbering (section, page, theorem, etc.) when citing Novak’s survey article
> - Corrected the GP prior input in Algorithm 2 (**$\mu_0$** is zero in our case)
> - Fix the typos mentioned

---

> > ### Comment · Reviewer_towK · 2023-05-05
> > **-**
> >
> > Thank you for your response. I would just note that while it is indeed not technically incorrect to state that the inequalities in Lemma 16 hold with probability one, to do so makes the claim slightly weaker than it could be: The inequalities hold not only almost surely but, in fact, surely. [Also, I guess there is a typo in the new version of the lemma: the inequalities should probably hold for any $x \in D$, not just for any $x \in X$.]

---

> > > ### Author Response · Authors · 2023-05-30
> > > **Thanks**
> > >
> > > Thank you for the follow-up comments.  We have fixed the typo in the lemma.

---

### Review · Reviewer_jwFj · 2023-04-07

**Summary Of Contributions:**

The paper provides several theoretical analyses for the error bounds of Bayesian Quadrature (BQ) in the noisy settings, randomized algorithms, and average-case performance measures. In particular, the paper introduces a two-step meta-algorithm that acts as a means to derive the average-case quadrature error with the L2-function approximation error. The theoretical analyses can be applied to different types of kernels including the Square Exponential and the Matern kernels. The paper also conducts some experiments in order to verify the behaviours of their algorithm.

**Audience:**

Yes

**Claims And Evidence:**

Yes

**Requested Changes:**

+ Please answer my concerns in the Weaknesses section.
+ [minor] The plots in Figs. 1 and 2 are a bit small and the font sizes are also a bit small. Maybe it’s better to increase the sizes to ensure readability.


**Strengths And Weaknesses:**

Strengths:
+ The paper is well-written; all the formulas and the theorems are well-described. Insights of the theorems are also well-explained in the paper.
+ I think the paper targets an interesting problem. It seems to me that the theoretical analyses are meaningful as understanding the error bounds of BQ is important. Although I can’t dig into detail to confirm the correctness of the theoretical analyses as I’m not very familiar with the theoretical analyses of BQ.
+ The paper also tries to evaluate their algorithm to see if it is effective in practice.

Weaknesses:
+ I have one concern regarding the experiments. I don’t understand why to obtain better understanding of the method, in the experiments, the paper modifies to alternate between maximum variance sampling and Monte Carlo sampling? Isn’t this result in a different method? I see the justification of “mathematically, this does not change the behavior of the final step” is not convincing. Why don’t we run experiments with the proposed algorithm? And why alternating will not change the behavior of the proposed method? And is this invariant behaviour only for the final step or for all the iterations? I think more rigorous justifications and explanations need to be provided. Ideally, as a reader I would expect to see the performance of the original proposed algorithm – the algorithm that is used to conduct the theoretical analysis.

---

> ### Author Response · Authors · 2023-04-27
> **Response to Reviewer jwFj:**
>
> Thank you for your feedback and the helpful review – we respond to the main points as follows.
>
> Q. **Why alternate between MVS and MC, will it change the final behavior of Algorithm 1?**
>
> A.  The final behavior is not affected at the *final* time: For example, if we have 200 time steps in total (say), then as long as we spend 100 time steps on MVS and 100 time steps on MC, the behavior of the error at time T=200 is exactly the same mathematically.  On the other hand, the behavior at the *intermediate* times would be affected.  The advantage of alternating between the two is that even if we look at the error at some intermediate time (e.g., T=100), we can be sure that the algorithm has allocated an equal budget to each of MVS and MC.
>
> Q. **The font size in Figs. 1 and 2 are not easy to see.**
>
> A.  Thanks for the suggestion; we have adjusted the font size in the plots to make them more readable.

---

### Review · Reviewer_s6oR · 2023-04-24

**Summary Of Contributions:**

This article studies error bounds for kernel-based quadrature rules. These algorithms are widely used in the literature of numerical integration. The focus was put on the noisy settings where the algorithm is randomised, or the integrand is noisy. In particular, a meta-algorithm was proposed that build a quadrature rule using an L2-function approximation of the integrand.

**Audience:**

Yes

**Claims And Evidence:**

Yes

**Requested Changes:**

I believe that a major revision is necessary in order to include a comparison of the contributions of this work to the existing work on the literature of control functionals.

**Strengths And Weaknesses:**

Both content and the presentation of this article fail to meet the standards of the journal. Indeed, the article lacks a paragraph, that follows the introduction, where the content of each section is described. In my opinion, the presentation may be challenging . For instance, the word ‘quadrature’ appears on the first page, while the first example of a quadrature rule appears on the sixth page. Morevover, it seems that the authors are not aware of the literature on control functionals. See, for instance, [1], where a variant of the meta-algorithm was proposed and analysed theoretically.


[1] Oates, Chris J., Mark Girolami, and Nicolas Chopin. "Control functionals for Monte Carlo integration." Journal of the Royal Statistical Society. Series B (Statistical Methodology) (2017): 695-718.

---

> ### Author Response · Authors · 2023-04-27
> **Response to Reviewer s6oR:**
>
> Thank you for your feedback and the helpful review – we respond to the main points as follows.
>
> Q. **The article lacks a paragraph, that follows the introduction, where the content of each section is described**
>
> A.  Section 1.1 was meant to serve this purpose, but we omitted the section numbers.  We have now listed such numbers at the end of Section 1.1.
>
> Q. **In my opinion, the presentation may be challenging. For instance, the word ‘quadrature’ appears on the first page, while the first example of a quadrature rule appears on the sixth page.**
>
> A.  We believe that our problem setup is outlined adequately around Eq. (1) and explained in detail adequately in Section 2.  If it is deemed necessary by the reviewer, we can replace the word “quadrature” by “[noisy] integration” when referring to the problem that we study.  We are also open to other suggestions, but we currently don’t see any major problems in terms of accessibility and clarity.
>
> Q. **Moreover, it seems that the authors are not aware of the literature on control functionals. See, for instance, [1], where a variant of the meta-algorithm was proposed and analysed theoretically.**
>
> A.  Thank you very much for pointing this out.  We have added a detailed discussion in Section 1.2, and also mentioned it briefly in Section 1.1, citing Oates et al. (2017) and some other works we believe to be representative of this literature.
>
> In short, we summarize the discussion as follows:
> - In terms of the general meta-algorithm approach, we had already highlighted its existence (citing the survey of Novak, who in turn cites Bakhvalov (1959)), and accordingly, we don’t believe that its use in this more recent work amounts to a substantial omission due to not having been cited.
> - In terms of the specific applications, we discuss in Section 1.2 how these appear to be very different, e.g., with the paper of Oates et al. adopting a set of technical assumptions quite distinct from ours and leading to a $T^{-7/6}$ convergence rate, which is sometimes unattainable and sometimes highly suboptimal the settings we focus on (e.g., noiseless Matern).  We also emphasize our focus on the noisy setting, whereas Oates et al. (2017) focused on noiseless.

---

> > ### Comment · Reviewer_s6oR · 2023-05-28
> > **-**
> >
> > Thank you for your response. It would be interesting to discuss the fundamental differences between the proof of your result and the proof of its equivalent counterpart in Oates et al. (2017). Also, I suggest using a log scale in Figure 1&2 to show the rates of convergence.

---

> > > ### Author Response · Authors · 2023-05-30
> > > **Further Response to Reviewer s6oR**
> > >
> > > Thank you for the additional comments – please see below for responses.
> > >
> > > Q. Highlight the proof differences between Oates et al.’s Proposition 1 and our Theorem 1.
> > >
> > > A. There are similarities in the two proofs, and also in earlier works that we highlight in the paper (as early as Bakhvalov (1959) as discussed following (3)).  Both consider variance in the residual estimate and use the unbiased property of the estimate.  On the other hand, we note the following:
> > > * Proposition 1 of Oates et al. is stated directly in terms of the variance of *$f(x_t) - \hat{f}(x_t)$* when *$x_t \sim p$* (in our notation), and thus roughly amounts to stopping at (46) in our Appendix E.  In contrast, we perform further steps to attain bounds in terms of the L2-function norm, to facilitate applying L2 bounds from earlier works.
> > > * Importantly, we also need to consider the impact of noise, and even our analysis leading to (46) has some extra effort for doing so compared to the noiseless setting of Oates et al.
> > > * As we discuss at the bottom of page 3, our work and theirs deviate considerably after having established the original meta-theorem.
> > >
> > > Q. Logarithmic scale to show the convergence.
> > >
> > > A. Thank you for the suggestion.  We have updated the manuscript to use a logarithmic y-scale whenever the x-axies is the number of queries (so the curves usually approach zero).  For the plots where “split fraction” is the x-axis, the values are generally further from zero and there is no “convergence” to be seen, so we still use a linear scale.

---

### Author Response · Authors · 2023-04-27
**Common Response**

We are very grateful to all reviewers for their helpful and insightful feedback, and to the Action Editor for handling our paper.  The responses are given individually to each reviewer, and the updated manuscript has the main changes highlighted in blue.  We believe that the paper has improved thanks to the reviewers, and that it now satisfies the TMLR criteria of being (i) correct, and (ii) of interest to some members of the ML community.

---

### Decision · Action_Editors · 2023-06-13

**Recommendation:** Accept as is

**Comment:**

Two of the three reviewers identified strengths of this paper, with one being more positive.

The key concern of Reviewer jwFj was in the way the experiments were run (by alternating between a maximum variance sampling (MVS) phase and a Monte Carlo (MC) sampling phase). From my look at the paper, I believe the authors’ choice here was justified, as both sampling methods (as mentioned by the authors in the paper) are non-adaptive. Hence, for any time point corresponding to the last time within a MC sampling phase, the distribution of the sample collected thus far will be the same as long as an equal number of rounds were allocated to each of MVS and MC. Therefore, I believe this concern is resolved.

The authors give both upper bounds and lower bounds, which I discuss in turn.

**Upper bounds**

As far as I am aware, the results for the squared exponential (SE) kernel in the noisy setting are new. For the Matérn kernel, the authors show that a two-phase sampling algorithm can recover the rate that Plaskota (1996) previously showed for another method. Plaskota's method involves repeatedly resampling at the same points in a specific set of points. The authors' method instead is unlikely to ever sample the same point twice. To consider why the author's method may be advantageous, let us consider active learning. In active learning methods, it is sometimes considered unrealistic to be able to request labels at the same point multiple times (this may be akin to asking a person to repeatedly label the same image multiple times). I do think it is a nice advantage that the authors' method therefore avoids repeated sampling, while still avoiding adaptive sampling.

**Lower bounds**

Next, I briefly discuss the lower bound. While the lower bound rate the authors' achieve is not new, the way in which they achieve this lower bound, via a single hard set of functions to get both terms in the lower bound, appears to be new. This adds sufficient interest to the lower bound and may be useful in future work.

In total, I believe the revised version is suitable for publication at TMLR.

**Audience:**

Overall, this work makes theoretical contributions that should be of sufficient to interest to those interested in numerical integration under noisy function evaluations. A potential concern is, as noted by Reviewer towK, that the work may be less interesting from a Bayesian quadrature perspective as the algorithm considered is not one that is typically used for Bayesian quadrature (and, empirically, the benefit is not clear).

**Claims And Evidence:**

Yes, the claims made are well-supported, and in the revised version the authors are clear about which results are new and about novelty in proofs versus novelty in rates.